# Toward grouped-reservoir computing: organic neuromorphic vertical transistor with distributed reservoir states for efficient recognition and prediction

Changsong Gao[1,2], Di Liu[1,2], Chenhui Xu[1,2], Weidong Xie[1,2], Xianghong Zhang [1,2], Junhua Bai[3], Zhixian Lin[1,4], Cheng Zhang[5], Yuanyuan Hu [6], Tailiang Guo[1,2] & Huipeng Chen [1,2] ✉

Reservoir computing has attracted considerable attention due to its low training cost. However, existing neuromorphic hardware, focusing mainly on shallow-reservoir computing, faces challenges in providing adequate spatial and temporal scales characteristic for effective computing. Here, we report an ultra-short channel organic neuromorphic vertical transistor with distributed reservoir states. The carrier dynamics used to map signals are enriched by coupled multivariate physics mechanisms, while the vertical architecture employed greatly increases the feedback intensity of the device. Consequently, the device as a reservoir, effectively mapping sequential signals into distributed reservoir state space with 1152 reservoir states, and the range ratio of temporal and spatial characteristics can simultaneously reach 2640 and 650, respectively. The grouped-reservoir computing based on the device can simultaneously adapt to different spatiotemporal task, achieving recognition accuracy over 94% and prediction correlation over 95%. This work proposes a new strategy for developing high-performance reservoir computing networks.

With the rapid development of artificial intelligence, the hardware artificial neural network (HW-ANN) technology, inspired by the human-brain, is seen as an effective solution to overcome the bottleneck of von Neumann architecture[1–5]. In recent years, HW-ANN has made major breakthroughs in applications such as pattern recognition[6], artificial vision[7] and cross-modal information processing[8], and so on, which has attracted great attention in the fields of intelligent driving, remote sensing imaging[9] and military industry. However, depending on the direction of the information flow in the neural network, feed-forward neural network (FNN) just allows signals to be passed from input to output, which is detrimental to the processing of spatiotemporal signal[10]. On the other hand, although recurrent neural network (RNN) have achieved excellent results in many tasks of processing spatio-temporal signals, the need for backpropagation through time (BPTT) algorithms to optimize recursive weights lead to slow convergence speed, difficulty in training, gradient vanishment/explosion and other problems[11]. Therefore, in recent years, the concept of reservoir computing (RC) with the ability to circumvent the problem of error

[1]Institute of Optoelectronic Display, National & Local United Engineering Lab of Flat Panel Display Technology, Fuzhou University, 350002 Fuzhou, China. [2]Fujian Science & Technology Innovation Laboratory for Optoelectronic Information of China, 350100 Fuzhou, China. [3]Joint School of National University of Singapore and Tianjin University, International Campus of Tianjin University, Binhai New City, 350207 Fuzhou, China. [4]School of Advanced Manufacturing, Fuzhou University, 362200 Quanzhou, China. [5]Department of Physics, Fuzhou University, 350108 Fuzhou, China. [6]Changsha Semiconductor Technology and Application Innovation Research Institute, College of Semiconductors (College of Integrated Circuits), Hunan University, 410082 Changsha, China. ✉e-mail: hpchen@fzu.edu.cn

accumulation in recursive networks has been proposed. Unlike traditional ANN techniques, only the weights connected to the output layer need to be trained in RC networks, and only extremely simple algorithms, such as linear regression, are required to perform recognition of input signals. Therefore, compared to traditional ANN, RC greatly reduces the training cost of the network[11–13] and attracts the attention of a large number of researchers[14–18].

Although a significant number of neuromorphic devices applied to RC have been reported in recent years, the majority of these efforts have focused on shallow-RC with monotonic reservoir state spaces[19]. This can be attributed to the heavy reliance on monotonic carrier dynamics when using reported neuromorphic devices as reservoirs to map sequence signals, which gives rise to several noteworthy issues for RC when performing different spatiotemporal tasks. One major issue is that the narrow range ratio of spatial characteristics makes it difficult to extract the diversity spatial feature of sequence signal, which greatly limits the richness of the reservoir space state. As a result, during the process of mapping complex sequence signals, the reservoir state tends to overlap, making it difficult to effectively separate the spatial characteristics within complex information and subsequently reducing recognition accuracy. Another issue is the limited rang ratio of temporal characteristic, which hinders efficient extraction of temporal feature from sequential signals with diverse time-scales. For example, when performing dynamic trajectory prediction with abundant time-scales, the limited range ratio of temporal characteristic is difficult to adapt to the signal with different temporal feature, which severely limit the correlation of prediction. Despite researchers have achieved multi-scale temporal characteristics by increasing the number of signal modes in the input layer based on shallow-RC networks[20], as shown in the Supplement Information Fig. S1, the limitation of shallow-RC on spatial characteristics remain unresolved. Furthermore, increasing the input layer also means the requirement of more encoding design for sequence signals and the utilization of more physical devices to receive different modes of physical signals. This significantly increases the signal error rate and pre-processing cost of the input signals, which is detrimental to the robustness of RC. Therefore, developing new neuromorphic reservoir devices along with new RC networks to simultaneously meet large-scale spatial and temporal characteristics are highly required, which is crucial for achieving high-performance recognition and prediction in complex spatiotemporal tasks for RC networks.

Interestingly, primates in nature are able to quickly and accurately recognize complex object information, such as facial recognition, with the help of advanced synaptic dynamics mechanisms. Brain science research on primates has confirmed[20–22] that primates use a distributed memory characteristic for processing complex information. When the nervous system processes a task, each neuron and neural circuit processes only a part of the information and generates a part of the output. For example, as shown in Fig. 1a, when a primate observes an unfamiliar face, neurons in the temporal polar (TP) region (blue) respond to familiar eye features, forming TP feature memory. Neuron cells in the anterior-medial (AM) region respond to unfamiliar lip features, forming AM feature memory[23]. In this way, all outputs are integrated by the cerebral cortex to form the final output result, significantly improving the computational efficiency and accuracy for complex information processing. The physiological significance of distributed memory characteristics in primates serves as inspiration for the design of physical node devices with distributed reservoir states in the reservoir layer of the RC system. These devices are intended to facilitate the distributed mapping of spatiotemporal signals. However, to date, no such devices have been demonstrated.

In this work, inspired by the distributed memory characteristic of primates, an ultra-short channel organic neuromorphic vertical field effect transistor with distributed reservoir states is proposed and used to implement grouped-RC networks. By coupling multivariate physical

mechanisms into a single device, the dynamic states of carriers are greatly enriched. As reservoir nodes, sequential signals can be mapped to a distributed reservoir state space by various carrier dynamics, rather than by monotonic carrier dynamics. Additionally, a vertical architecture with ultra-short nanometers transport distance is adopted to eliminate the driving force of the dissociation exciton, thereby improving the feedback strength of the device and the reducing the overlap between different reservoir state space, which only cause negligible additional power. Consequently, the device serves as a reservoir capable of mapping sequential signals into distributed reservoir state space with 1152 reservoir states, and the range ratio of temporal (key parameters for prediction) and spatial characteristics (key parameters for recognition) can simultaneously reach 2640 and 650, respectively, which are superior to the reported neuromorphic devices. Therefore, the grouped-RC network implemented based on the device can simultaneously meet the requirements of two different spatiotemporal types task (broad-spectrum image recognition and dynamic trajectory prediction) and exhibits over 94% recognition accuracy and over 95% prediction correlation, respectively. This work proposes a strategy for developing neural hardware for complex reservoir computing networks and has great potential in the development of a new generation of artificial neuromorphic hardware and brain-like computing.

## Result
### Grouped-RC and device design

The face recognition process in primates, as shown in Fig. 1a, involves a unique memory mechanism of distributed processing in synaptic dynamics. When the monkey brain receives facial information $S_1$ from each other, nerve cells and neural circuits in different regions of the brain process the feature information in $S_1$ separately, and obtain different spatial-temporal feature memories, such as $N_1$, $N_2$, $N_3$, and so on. Eventually, the cerebral cortex fuses these memories and makes judgments. Given the efficient distributed memory processing of the primate brain, we expect to introduce this physiological mechanism into RC systems so that the system has richer reservoir state space. Therefore, as shown in Fig. 1b, inspired by the distributed memory characteristic of monkeys, the reservoirs in the RC system process the input signal in parallel based on different carrier dynamics (such as marked yellow, purple, blue, and red). The final input signal is mapped to the reservoir state space of different dimensions, allowing the system to obtain a wide range ratio of spatial-temporal characteristics ratios, i.e. grouped-reservoir computing. However, achieving this process at the physical device level is a challenge, as it requires reservoir devices to possess device attributes of non-linear response characteristics and short-term memory characteristics, while also needing a wide dynamic range of feedback intensity and time characteristics to meet the demands of a large number of reservoir states. (details are discussed in Supplementary Information Note 1). Although the use of dynamic memristors has been widely reported, its limited number of terminals can easily cause the reservoir to become a relatively fixed nonlinear function[24]. At the same time, the limitations of the photogenerated charge transport efficiency due to the long transport distance of conventional transistors can easily lead to a narrow range of feedback intensities F. Here, we propose an organic neuromorphic vertical field effect transistor with distributed reservoir states (VOFET-DR) as the reservoir, whose structure is shown in Fig. 1c and Supplementary Information Fig. S2. In particular, we achieve large-scale $\tau$ using organic semiconductor materials with broad spectral absorption characteristics and couple it with the vertical architecture to broaden the F range of the reservoir.

An organic semiconductor layer consisting of a bulk heterojunction (BHJ) of N2200(poly{[N,N′-bis(2-octyldodecyl)-naphthalene-1,4,5,8-bis(dicarboximide)−2,6-diyl]-alt-5,5′-(2,2′-bithiophene)]}) (n-type): POFDIID (conjugated polymers of fluorinated iso-indigo [7,6-g]

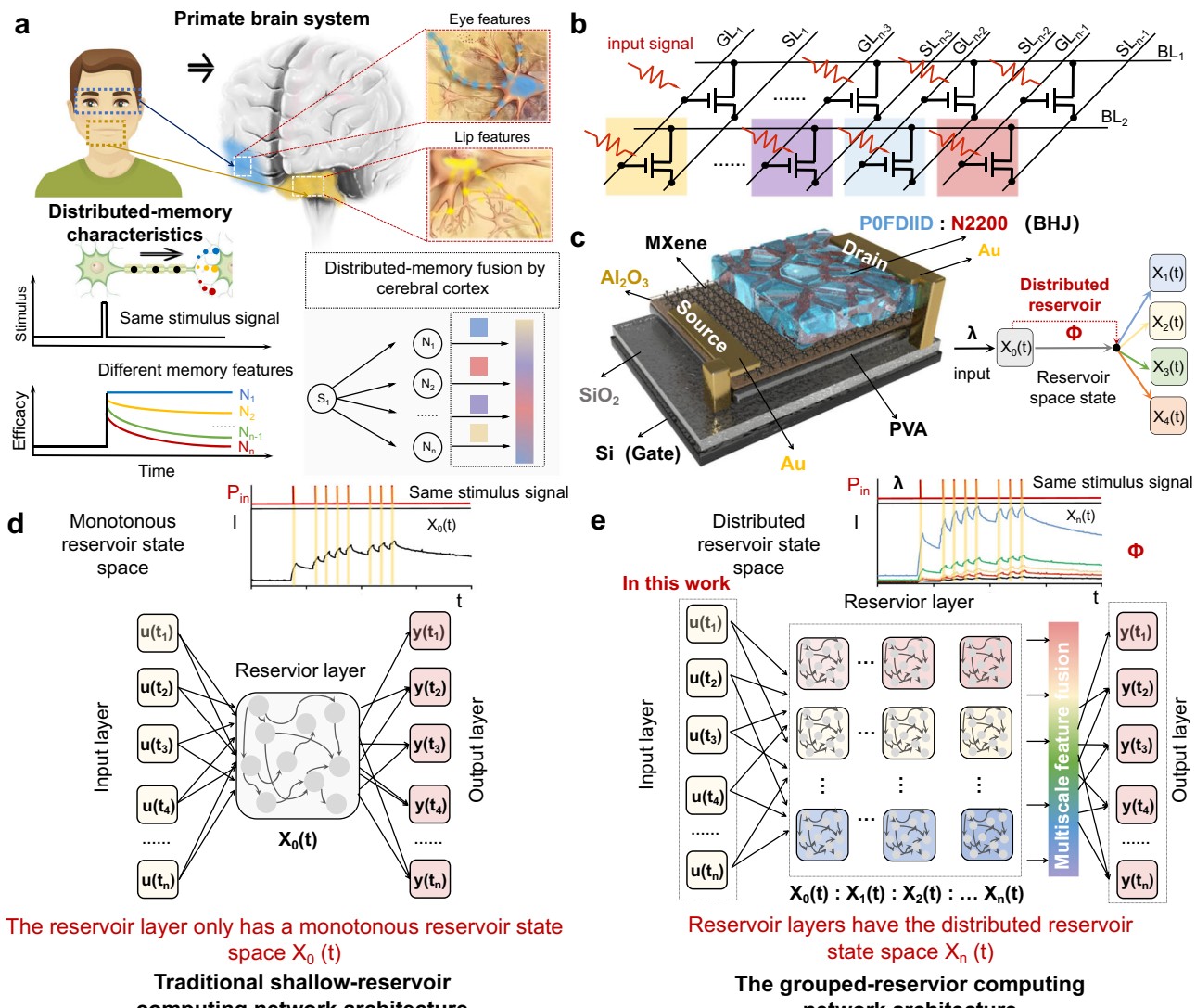

**Fig. 1 | Distributed memory processing of primates and grouped-reservoir computing based on VOFET-DR. a** The distributed memory processing characteristics in primates. Nerve cells located in different brain regions perform specific processing and memory on a part of the input information, and finally integrate and judge all the processed information through the cerebral cortex. **b** VOFET-DR with different temporal characteristics (marked by different colors) as nodes in the reservoir. **c** Structure diagram of VOFET-DR. The device can adaptively adjust different temporal characteristics and feedback strength according to the wavelength of the input optical signal or the applied gate voltage. **d** Schematic diagram of network architecture for traditional shallow-reservoir computing. **e** Schematic diagram of network architecture for grouped-reservoir computing.

iso-indigo) (p-type) is used as the active layer. In particular, N2200 is used as an electron acceptor for charge trapping, while P0FDIID with broad-spectrum absorption properties is used as a light absorbing layer and the main channel material to ensure efficient charge transport. The absorption spectra, as well as the chemical structures of the active layer materials, are shown in Supplementary Information Figs. S3 and S4, respectively. Since photogenerated charge transfer[25] and trapping effects[26–28] occur readily between p-type and n-type organic semiconductor interfaces, when photons with different energy are fed into the device, the device is able to generate short-term memory photocurrents with different temporal characteristics to capture different features of the physical signal. Due to the nanoscale channel length of the adopted vertical field effect transistor structure (the channel length is the thickness of an active layer, which is ~65 nm, as shown in the Supplementary Information Fig. S5), it provides an uneven and large electric field that greatly promotes the separation and transport of photogenerated carriers[29–31]. This effectively reduces non-radiative recombination and improves the device feedback intensity, which provides a wide range of spatial characteristics. The

network source electrode of the vertical transistor is composed of MXene thin film containing perforations to avoid the shielding effect of the gate electric field from the source[32]. In Supplementary Information Figs. S6–S8, MXene films are further characterized by scanning electron microscopy (SEM), the X-ray photoelectron spectroscopy (XPS) spectra and atomic force microscopy (AFM), respectively. $Al_2O_3$ (1 nm)/PVA (polyvinyl alcohol) is used as a charge-trapping layer to trap dark-state carriers to avoid excessive off-state currents. The specific fabrication process of the device is described in detail in the experimental method.

In conventional shallow-RC[10,19,24,33–36], as shown in Fig. 1d, physical node devices in the reservoir layer map the inputting sequence signal based on relatively simple carrier dynamics, resulting in monotonous reservoir state space, which is defined as $X_0(t)$. This greatly limits the range ratio of spatial and temporal characteristics of RC. In this work, as shown in Fig. 1e, we utilize the VOFET-DR as a single physical node in the reservoir layer, enabling the design of grouped-RC. The device is capable of sensing optical sequence signals with different wavelengths and generating memory currents, thereby allowing the nonlinear

temporal characteristics of the sequence signals to be mapped into the reservoir space $X_0(t)$. Due to the varying single-photon energies associated with input light sequence signals of different wavelengths, the resulting memory current exhibits distinct temporal characteristics depending on the wavelength. Consequently, the physical node can first map the input optical sequence signal to different reservoir spaces $X_1(t)$, $X_2(t)$, $X_3(t)$, $X_4(t)$, and so on, based on the specific wavelength. Additionally, the vertical field-effect transistor has the capability to manipulate the Schottky barrier between the active layer and the source interface through gate bias, which allows the device to adjust the charge injection and overall current of the device, resulting in memory currents with different spatial characteristics. By further setting the gate bias of the device, it becomes possible to map different spatial characteristics $X_1(t)$, $X_2(t)$, $X_3(t)$, $X_4(t)$, and so on, based on the original spatial characteristic $X_0(t)$. As a result, the physical nodes can map different spatial-temporal characteristics based on different carrier dynamics, effectively meeting the requirements of grouped RC.

### Field effect characteristics of the device

Figure 2a shows the operating mode of VOFET-DR as a physical node in grouped-RC. The device utilizes vertical field effect transistor architecture and a p-n organic semiconductor bulk heterojunction (BHJ) as the active layer. Thus, the device has three modes of operation to map the nonlinear temporal characteristics of the sequence signal to reservoir space. The first mode is inputting voltage sequence pulse signals to the device. The second mode is inputting laser sequence pulse signals to the device. The third mode is simultaneously applying gate bias while inputting laser sequence pulse signals to the device. As the field-effect mechanism of the vertical transistor can modulate the efficiency of photogenerated charge separation in bulk heterojunction and facilitate the injection of source charge[29,32], the third mode is able to map the temporal characteristics of light signal into different reservoir spaces. To verify the viability of this strategy, the field effect transistor properties of the device are initially investigated. Figure 2b illustrates the transfer curve, suggesting that the gate bias $V_{GS}$ can effectively regulate the output current of the transistor. Next, Fig. 2c shows the variation of the hysteresis window after the device is applied with different double sweep voltages $V_{GS}$, which shows that the hysteresis window increases with the $V_{GS}$ sweep range, indicating that the device has a memory effect and has the potential to be used as a reservoir[34,36].

### Nonlinear response and short-term memory characteristics of the device

Therefore, to investigate the ability of the device to act as a reservoir physical node, the device is subjected to single $V_{pulse\ GS}$ with different amplitude. The device demonstrates the short-term memory currents (see Fig. 2d and Supplementary Information Fig. S9), which satisfy the requirement of a reservoir physical node to map a sequence signal. In addition to the input $V_{pulse\ GS}$, the devices are individually applied with a single light pulse signal of different wavelengths (no gate voltage) and similarly exhibit short-term memory current, as shown in Fig. 2e and Supplementary Information Fig. S10. To further analyze the impact of input signals with different modes on performance, the nonlinear temporal characteristic τ of the device is extracted in both operating modes. This parameter is crucial for evaluating the ability to map the sequence signal, as illustrated in Fig. 2f. The method of extraction of nonlinear temporal characteristics τ is elucidated in the Supplementary Information Note 2. It can be found that for the $V_{pulse\ GS}$ mode, the device has a relatively narrow range of nonlinear temporal characteristics (from 0.14 s to 0.39 s), which is much lower than that for the mode of light pulses (from 0.005 s to 1.72 s). In RC, the computing capacity of the system depends primarily on the physical nodes in the reservoir capturing different temporal characteristics and mapping these characteristics to the reservoir space[11,19]. Therefore, having a wide

range ratio of temporal characteristics is essential to enhance the reservoir state richness of the system. Given that the light pulse mode has wider range of nonlinear temporal characteristics, using the light pulse as the input signal for this device is more conducive to the design of grouped-RC.

On the other hand, due to the point-wise separation property for the reservoir[33], the feedback strength of the physical device has a critical impact on the spatial characteristics of reservoir state. For the $V_{pulse\ GS}$ mode, $F_E$ is defined to evaluate the feedback strength. The equation is as follows:

$$F_E = \frac{\Delta P_{spike\ out}}{P_{write}} \tag{1}$$

where $\Delta P_{spike\ out}$ is the variable of output spike power, $P_{in}$ is the input power of the write pulse. For the light pulse mode, we define $F_{ph}$ to evaluate the feedback strength. The equation is as follows:

$$F_{ph} = \frac{\Delta I_{spike\ out}}{P_{light\ in}} \tag{2}$$

where $\Delta I_{spike\ out}$ is the variable of output spike current, and $P_{light\ in}$ is the light power density of the input light pulse. Figure 2g shows the feedback strength of the $V_{pulse\ GS}$ mode (top) and the light pulse mode (bottom), respectively. Standard deviation (SD) was used to assess the range of variation in feedback strength. The method of computing SD is elucidated in the Supplementary Information Note 3. It can be found that the SD of the $V_{pulse\ GS}$ mode is 15.5, implying a relatively constant variation in feedback strength, in contrast to the SD of 280.6 for the light pulse mode, indicating a rich variation in feedback strength. This suggests that the light pulse mode allows the reservoir to extract spatial characteristics in the sequence signal more efficiently and to generate diverse short-term memory dynamics. Considering that objects in nature possess different electromagnetic spectral properties and reflect electromagnetic wave signals of varying wavelengths, such as the 808 nm band highlighting the information of "soil" and "trees," and the 450-650 nm visible band highlighting "highway" and "water," and that the device is capable of generating rich short-term memory dynamics for light pulses of different wavelengths, light pulses is chosen as the carrier of sequence signals for the device to extract the feature information of the target more efficiently and to improve the computing capacity of RC.

### Nonlinear mapping of multi-bit signals

In addition to rich short-term memory characteristic dynamics, efficient mapping of spatiotemporal characteristics of the sequence signal into the reservoir space is essential for RC[11]. To evaluate the mapping capabilities, a 6-bit light sequence signal test is performed by randomly input four types of sequence light pulse signals, as shown in Fig. 2h. Each periodical input waveform (650 nm, 0.01 mW cm$^{-2}$, 0.1 s pulse width, 0.2 s pulse interval) is considered as one bit, in which the light pulses "on" and "off" represent the "1" and "0" in the binary code respectively. It is obvious that as the state $x(t_n)$ of the device is influenced by the input state $u(t_n)$ in conjunction with the input state $u(t_{n-1})$ at the previous moment, different sequence signals such as '001010', '100011', '101011', '110101' are able to be mapped by the device with different magnitudes of current. For example, '001010', '100011', '101011', '110101' correspond to 2.9 nA, 3.6 nA, 6.13 nA, 4.32 nA respectively. Further, 64 types of binary timing signals from '000000' to '111111' are fed into the device, as shown in Fig. 2i and Supplementary Information Fig. S11. It demonstrates 64 different conductivity states, and the sample deviations of these conductance states are shown in Supplementary Information Fig. S12. This shows that the device can effectively map the nonlinear temporal characteristics of different

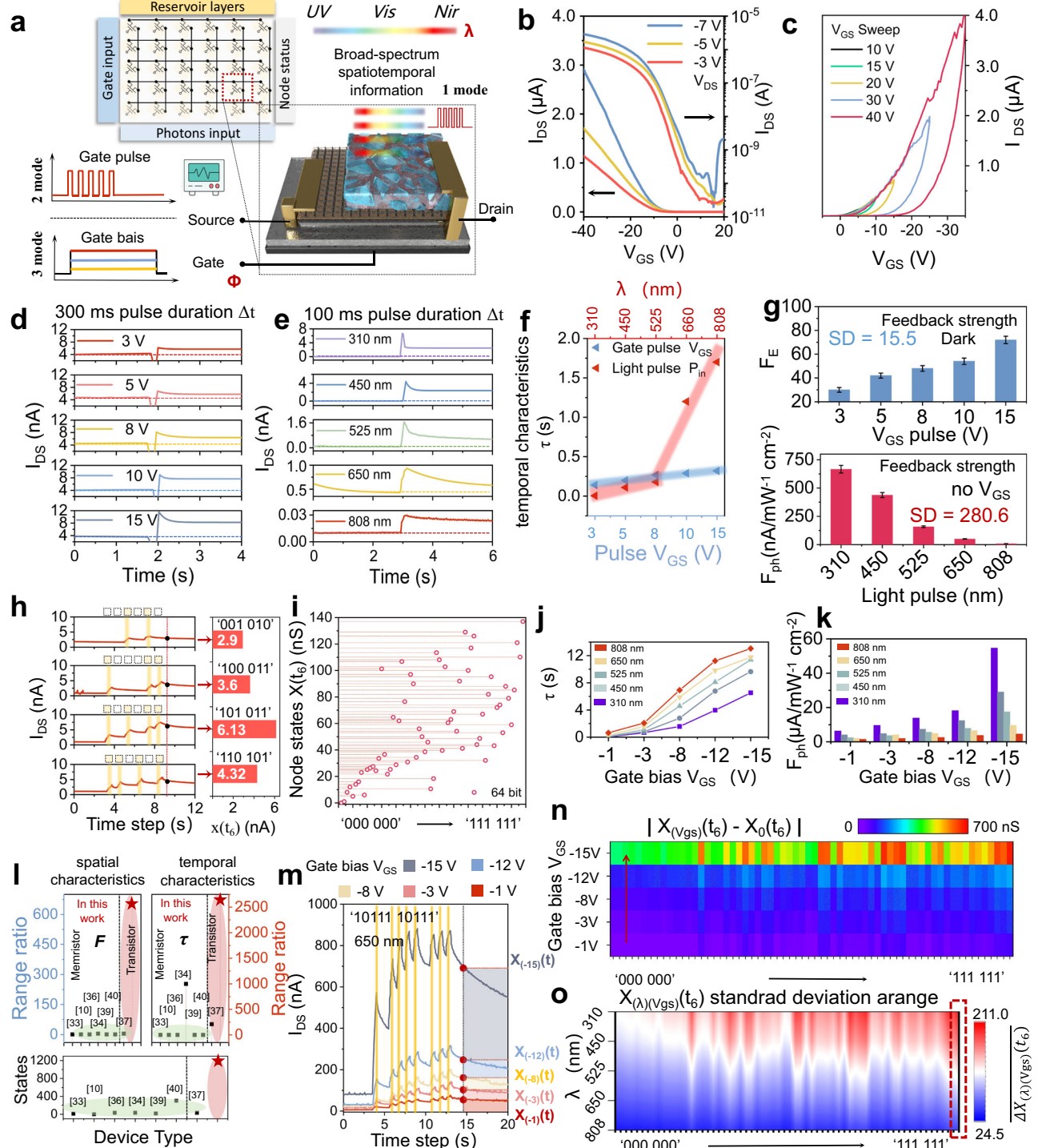

**Fig. 2 | Photoelectric performance of VOFET-DR. a** There are three modes of VOFET-DR as the physical node of the reservoir. **b** Transfer curve of VOFET-DR. **c** double-layer sweep transfer curves for different $V_{GS}$ ranges ($\Delta V_{GS}$ from 10 V to 40 V, $V_{DS}$ = −10V). **d** The output current of the device after being fed $V_{GS}$ pulses of different amplitudes (pulse width $\Delta t$ = 300 ms, $V_{DS}$ = −1 V). **e** The output current of the device after being fed light pulses of different wavelengths (pulse width $\Delta t$ = 100 ms, $V_{DS}$ = −1 V, light intensity $P_{in}$ = 0.01 mW/cm²). **f** Nonlinear temporal characteristics extracted from **d** and **e**. **g** The feedback strength of the input electrical pulse (top) and the input light pulse (bottom). **h** Reservoir state X ($t_6$) after the device is input with different optical pulse signals. **i** I$_{DS}$ responses generated by

optical pulse ranging from (000000) to (111111). **j** nonlinear temporal characteristics of a single light pulse (from 310 nm to 808 nm) input to the device are extracted under different $V_{GS}$ biases. **k** Optical feedback strength of the device under different $V_{GS}$ biases. **l** Performance comparison of related work. **m** At different $V_{GS}$ biases, the device is fed the output current after the optical sequence pulse (650 nm). **n** The difference between the resulting reservoir states after the device is input from (000000) to (111111) optical sequence pulses (650 nm) at different $V_{GS}$ biases. **o** The standard deviation of the reservoir state produced by the input of optical sequence pulses with different wavelengths to devices with different $V_{GS}$ biases.

sequence signals into the reservoir space, which is important for reservoir computing.

## Distributed reservoir states of VOFET-DR

In the primate brain, neuron, and neural circuit extract facial feature information of targets from different dimensions to accurately identify target identities. Thus, to construct this bionic processing mechanism, physical nodes in reservoirs map the spatiotemporal characteristics of sequential signals into reservoir spaces of different dimensions through different dynamics to achieve distributed reservoir states, which is essential for grouped RC. Hence, while the light pulse signal is input to the device, different gate bias $V_{GS}$ are applied to the device, expecting to further enrich the carrier dynamics, as shown in Supplementary Information Fig. S13. It can be observed that after adding the gate bias $V_{GS}$, the device outputs current at different amplitudes due to different feedback strength, which indicates that the input light pulse signals can be mapped to different reservoir state spaces.

In Fig. 2j, the nonlinear temporal characteristics of a single light pulse are extracted under different $V_{GS}$ biases. It can be observed that the device exhibits a wide range of temporal characteristics (ranging from 0.005 s to 13.2 s) when a $V_{GS}$ bias is added, which meets the requirement of RC for multi-scale temporal characteristics. However, an important consideration is the additional power caused by gate control, which is largely dependent on the gate leak current ($I_{gs}$). As shown in Supplemental Information Fig. S14, the $P_{gs}$ ($V_{GS} \times I_{gs}$) density is approximately $10^{-4}$–$10^{-3}$ (mJ s$^{-1}$cm$^{-2}$), which accounts for only 0.0004% of the $I_{ds}$ density 25 (mJ s$^{-1}$ cm$^{-2}$). At the same time, when an additional gatee voltage bias of -15 V is applied, the feedback intensity increases to 9.53($\mu$A mW$^{-1}$ cm$^{-2}$), which is a 190-fold increase compared to no gate (50 nA mW$^{-1}$ cm$^{-2}$). Therefore, the power derived from external electric filed could be negligible. Furthermore, in Fig. 2k, the feedback strength is calculated according to Eq. (2). It can be observed that the feedback strength of the physical node increases with $V_{GS}$ bias, For example, $F_{ph}$ increases from 6.3 $\mu$A/(mW cm$^{-2}$) to 13.8 $\mu$A/(mW cm$^{-2}$) to 54.6 $\mu$A/(mW cm$^{-2}$) for 310 nm pulses at $V_{GS}$ bias of -1V, -8V, -15V. This confirms that physical nodes with different biases $V_{GS}$ can obtain different degrees of memory effects from the same input signal, which is crucial for mapping the nonlinear temporal characteristics of sequence signals to reservoir state spaces of different dimensions. A wide range of temporal and spatial characteristics is key to further implementing complex-RC. Therefore, a range ratio of two parameters is introduced to evaluate the performance of the device as a reservoir node. As shown in Fig. 2l and Supplementary Information Table S1, the range ratios of temporal and spatial characteristics of VOFET-DR are shown as 2640 and 650, respectively, which are superior to currently reported neuromorphic devices for RC[10,11,33,35–38].

Furthermore, to examine the impact of $V_{GS}$ bias on the mapping of sequence signals, the device is subjected to a light pulse sequence of '1011110111' under varying $V_{GS}$ biases. As shown in Fig. 2m, it can be observed that different $V_{GS}$ biases enable the physical nodes to map the sequence signals into distinct reservoir state spaces. For example, $V_{GS}$ (-15V), $V_{GS}$ (-12V), $V_{GS}$ (-8V), $V_{GS}$ (-3V), and $V_{GS}$ (-1V) correspond to reservoir spaces $X_{(-15)}(t)$, $X_{(-12)}(t)$, $X_{(-8)}(t)$, $X_{(-3)}(t)$, $X_{(-1)}(t)$, respectively. The reservoir states of the device after 64 optical pulse sequences ranging from '000000' to '111111' are shown in Supplementary Information Fig. S15 under different bias $V_{GS}$ conditions, which shows 384 reservoir states. This proves that coupling photoconductivity and field effects can bring rich reservoir states to the reservoir. Similarly, the effect of wavelength on physical node mapping sequence signals is investigated in Supplementary Information Fig. S16. It is observed that, under the same gate bias ($V_{GS} = -10V$), sequential signals with different wavelengths can be mapped to different reservoir state spaces. For example, $P_{in}$ (310 nm), $P_{in}$ (450 nm),

$P_{in}$ (525 nm), $P_{in}$ (650 nm), and $P_{in}$ (808 nm) correspond to X $_{(310)}(t)$, X $_{(450)}(t)$, X $_{(525)}(t)$, X $_{(650)}(t)$, and X $_{(808)}(t)$ respectively. This confirms that VOFET-DR, as a physical node, can map the same sequence signal into the reservoir state space X(t) with different dimensions through different carrier dynamics to form different memory states, which is the distributed reservoir. Further, the optical pulse input signals in the three bands of ultraviolet (310 nm), visible (650 nm), and near-infrared (808 nm) light, ranging from '000000' to '111111', combined with different $V_{GS}$, can result in 1152 reservoir states, as shown in Supplementary Information Fig. S17.

In reservoir computing, the input signal, which that was challenging to divide in the low-dimensional space, can be linearly divided due to the enhanced distinction of sequence signal characteristics in the high-dimensional state space. Hence, the effect of bias $V_{GS}$ on the input signal of reservoir space segmentation is further evaluated. The device is fed a sequence of 64 optical pulse signals ranging from '000000' to '111111'. After 1 second from the end of input signal, the output current is defined as reservoir state X(t$_6$). As shown in Fig. 2n and Supplementary Information Table 2, by differentiating X(t$_6$) under different biases, it can be found that the value difference increases as bias $V_{GS}$ increases. This indicates that the bias $V_{GS}$ can effectively adjust the degree of state overlap between the reservoir state spaces $X_{(Vgs)}(t)$ of different dimensions, and enhances the linear segmentation of the input signal from the low-dimensional state space. Additionally, because sequential pulse signals with different wavelengths can also be mapped in different reservoir state spaces $X_{(\lambda)}(t)$, it brings richer dynamics of carrier to the reservoir. Therefore, different $V_{GS}$ and different wavelengths are combined to further separate the input signal. We input 64 sequences of light pulse signals from '000000' to '111111' at wavelengths of 310 nm, 450 nm, 525 nm, 650 nm, and 808 nm to VOFET-DR with different $V_{GS}$ biases, and the reservoir state $X_{(\lambda)(Vgs)}(t_6)$ was sampled in the same way. The degree of linear segmentation of the input signal is then evaluated by calculating the SD of the different reservoir states. Figure 2o and Supplementary Information Table 3a and 3b shows the $X_{(\lambda)(Vgs)}(t_6)$ standard deviation range based on the 64-type sequence light signals of different wavelengths. For example, for the '111111' sequence signal, the SD of the reservoir states of 310 nm, 450 nm, 525 nm, 650 nm, and 808 nm are 211, 179, 130.6, 89.1, and 44.2, respectively. The sequence signal with 310 nm exhibits a relatively large range. This demonstrates that the coupling of photon energy with different field effects allows for effective modulation of the degree of linear separability of input signals in low-dimensional state space. This demonstrates that by combining the wavelength $\lambda$ and the bias $V_{GS}$, the degree of linear segmentation of the low-dimensional state space of the input signal can be effectively adjusted, which is crucial for processing complex sequence signals.

## The working mechanism of VOFET-DR

To elucidate the mechanism underlying the distributed reservoir of the devices, the performance of devices that are not mixed with N2200 is investigated, as presented in Supporting Information Fig. S18. Compared with devices that used heterojunction as channels, devices without N2200 failed to exhibit memory characteristics under the stimulation of light pulses. This indicates that memory characteristics are related to bulk heterojunction structure. Hence, the mechanism of the device is analyzed and discussed in Fig. 3. The energy band structure of the material is shown in the Support Information Fig. S19. Initially, we investigated the case where the device is individually stimulated by light pulses. The bulk heterojunction structure is commonly used in organic photovoltaic cells to efficiently generate excitons that dissociate at the donor-acceptor interface, leading to the formation of photogenerated electron-hole pairs. Thus, when the light pulse is applied to the device, the same process of photogenerated electron-hole pair generation occurs in the channel, as shown in Fig. 3a.

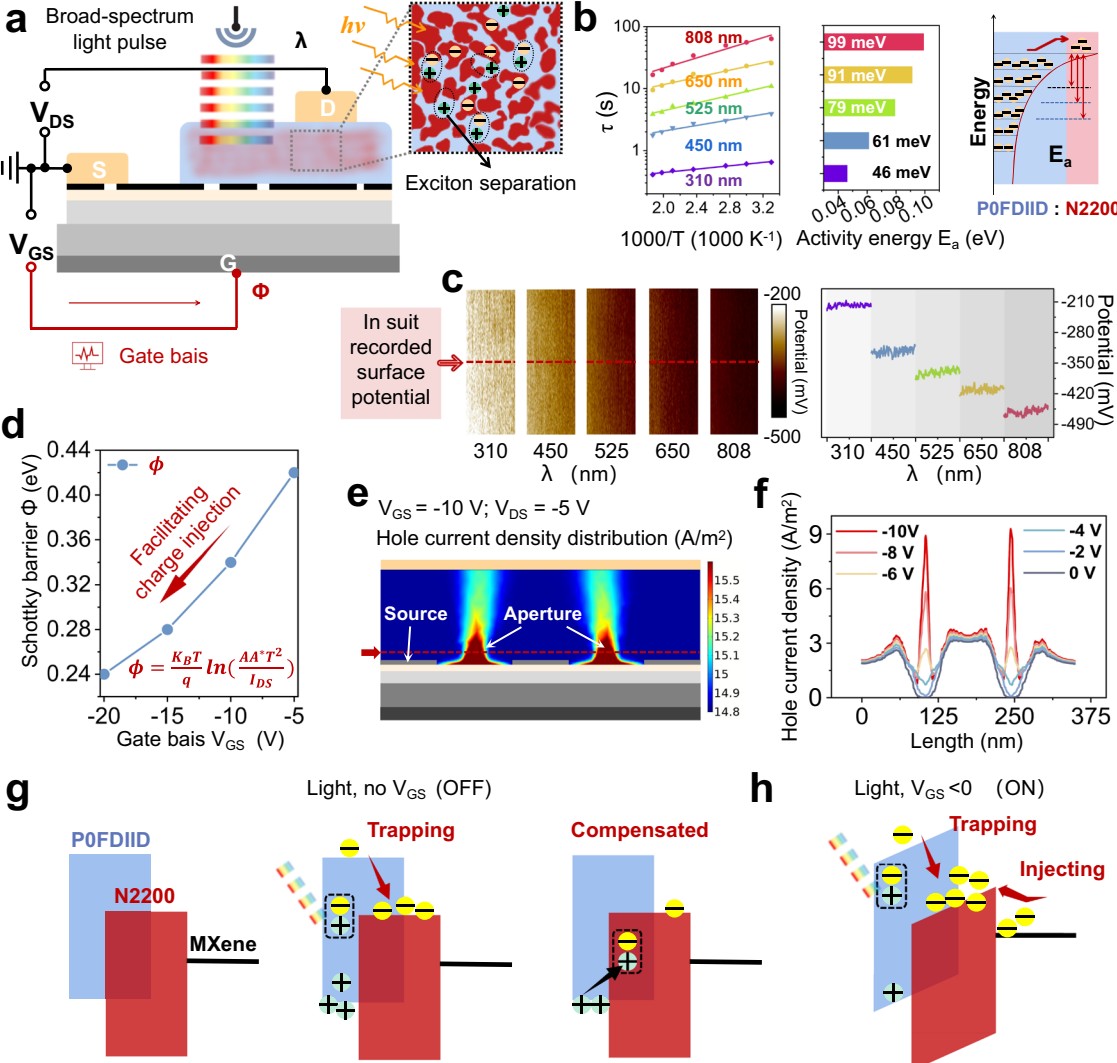

**Fig. 3 | Working mechanism of VOFET-DR. a** Schematic diagram of VOFET-DR operation. The device can independently receive a wide spectrum optical sequence pulse signal λ, and map its nonlinear temporal characteristics into the reservoir space. The gate bias can also be applied to the device to further adjust the dynamic state of the carrier by adjusting the height of the Schottky barrier of the vertical transistor. **b** The activation energy calculated according to Arrhenius law. **c** The surface potential of the mixed films was measured by KPFM under different wavelengths of light. **d** The variation of Schottky height under different gate voltage. **e** Current density distribution in the active layer calculated by COMSOL. **f** The current density extracted from the dashed line in **e** evolves with $V_{GS}$. **g** Schematic diagram of the dynamic changes of charge carriers after the device is input with light pulses when it is not added to $V_{GS}$. **h** Schematic diagram of the dynamic changes of charge carriers after the device is input with light pulses when added to $V_{GS}$.

Next, since the LUMO of P0FDIID is higher than the LUMO of N2200, the dispersed N2200 mixed into P0FDIID forms a potential barrier structure similar to a quantum well for charge trapping[27]. Moreover, the evolution of thin film morphology with and without the addition of N2200 is further investigated by atomic force microscopy (AFM), as presented in Supplementary Information Fig. S20. The dispersed N2200 results in higher roughness of the film, which means that more trapping sites are created.

Further, to investigate the impact of light pulses with different wavelengths on memory current, the activation energy $E_a$ for charge trapping under various wavelength inputs is calculated and shown in Fig. 3b (The calculation method is explained in Supplementary Information Note 4). It can be found that the $E_a$ corresponding to 310 nm, 450 nm, 525 nm, 650 nm, and 808 nm are 46 meV, 61 meV, 79 meV, 91 meV, and 99 meV, respectively. As the electron trapping is impeded by a higher energy barrier[39–41], the resulting memory current response is lower. Therefore, by extracting the memory current at 310 nm, which corresponds to a relatively minimum energy barrier 46 meV, a larger

memory current is obtained. This finding is consistent with the results presented in Fig. 2g. The error bar range is derived from the maximum and minimum values after 5 samples of experimental data. To further confirm that the wavelength can affect the electron trapping in different degrees, the surface potential distribution of the P0FDIID:N2200 mixed film was probed using kelvin probe force microscopy (KPFM) at different wavelengths of light source irradiation, as shown in Fig. 3c. It can be found that the surface potential of the mixed film decreases with wavelength, which is attributed to the increased electron concentration[42]. Therefore, the KPFM results further confirm the different effects of wavelengths on the electron trapping effect, which is consistent with the above results.

Another key to the distributed reservoir states is the effect of gate bias $V_{GS}$. The modulation effect of $V_{GS}$ on the device is discussed. For vertical field-effect transistors, the gate electric field affect the injection and transport state of carriers, which can be quantitatively analyzed by the test of temperature-dependent output characteristics, which are shown in Supplementary Information Fig. S21. The Schottky

current $I_S$ can be described utilizing thermionic emission model following the function[43] as:

$$I_S = AA^*T^2 \exp(-q\phi/k_B T) \tag{3}$$

in which $k_B$, $T$, and q are the Boltzmann constant, absolute temperature, and elementary charge, respectively. $\phi$ is the Schottky barrier height of the interface between MXene and the active layer, $A$ is the area of Schottky contact, and $A^*$ is the effective Richardson constant. $\phi$ is obtained by the slope, as shown in Fig. 3d. It can be seen that the $\phi$ decreases as $V_{GS}$ increases, indicating that $V_{GS}$ can effectively adjust the interface potential barrier to control charge injection. To further investigate the influence of $V_{GS}$ on charge injection, we analyze the potential and charge distribution within the device using COMSOL semiconductor device emulation calculation (Supplementary Information Table 4). Supplementary Information Figs. S22–S24 shows the potential distribution of the device at different $V_{GS}$. For $V_{GS} < 0$, a potential gradient is formed between the source and the aperture, facilitating charge injection. An analysis of the influence of the $V_{GS}$ on the potential distribution at the dotted line reveals that the potential within the aperture undergoes significant changes with varying $V_{GS}$, far exceeding the area outside the aperture. This demonstrates that the $V_{GS}$ can effectively adjust the electric field distribution inside the device, which is crucial for regulating the carrier transport state. Further, the influence of $V_{GS}$ on the charge distribution in the device is analyzed in Fig. 3e and Supplementary Information Fig. S25. When $V_{GS} < 0$, the charge in the aperture accumulates to form a 'virtual contact'[44,45], as shown in the Supplementary Information Fig. S25a–c, indicating that $V_{GS}$ greatly affects the charge injection effect. Moreover, Fig. 3f shows that the influence of the $V_{GS}$ on the charge distribution at the dotted line, and shows that a large amount of charge aggregation occurs in the aperture area as the $V_{GS}$ increases. This further confirms that $V_{GS}$ can effectively regulate the charge injection into the active layer and affect the distribution of the charge in the channel, resulting in different levels of output currents from the device.

Therefore, based on the above results, the working mechanism of the device is explained as follows: When the device is not applied with $V_{GS}$, as shown in Fig. 3g. It absorbs photons and generates photogenerated excitons when exposed to light pulse. Because P0FDIID is mixed with a small amount of N2200, the dispersed N2200 and P0FDIID form quantum well-like trapping sites (P0FDIID / N2200 / P0FDIID). When the photogenerated excitons separate into electrons and holes, the electrons are trapped by dispersed N2200, leading to a higher hole concentration in the channel P0FDIID and increased output current. After the light pulse ends, the trapped electrons in N2200 are compensated by the holes in the channel, resulting in a gradual decrease in the output current. The result is the phenomenon of short-term memory current. Furthermore, since the electrons photogenerated by light pulses with different wavelengths are hindered by energy barriers of different level during the process of charge trapping, which enables the device to effectively capture different physical characteristics of external information. When the device is applied with $V_{GS}$, as shown in Fig. 3h, holes accumulate at the interface between the semiconductor and the insulating layer. This causes the energy level of the semiconductor to bend, reducing the Schottky barrier $\phi$ between the source and the semiconductor. As a result, the electric filed induced by gate voltage prompts that excitons can be efficiently dissociated with a small driving force. Meanwhile, the electric filed contributes to the dissociation of charge-transfer state excitons, decreasing the non-radiative recombination and improve the feedback strength. Therefore, depending on the gate bias, different concentrations of charges are injected into the active layer, enriching the carrier dynamics. The optical sequence signal can be mapped to different reservoir state spaces by different carrier dynamics, so that its spatiotemporal characteristics can be effectively separated and distributed reservoir states can be realized.

## Grouped-RC for satellite remote sensing image recognition

High-precision satellite remote sensing image recognition is a challenge due to the complex spectral information contained. In view of the distributed reservoir states, we construct reservoirs based on VOFET-DR to separate the physical properties of sequence signals in multi-dimensions, and propose grouped-RC to identify complex features of ground objects in satellite remote sensing images.

As shown in Fig. 4a, different areas of the landform can reflect or radiate electromagnetic waves of different wavelengths. For example, desert and rocky areas can reflect more infrared electromagnetic waves, mountainous areas are prone to reflect a large number of ultraviolet electromagnetic waves, and urban building areas can reflect more visible light waves. Therefore, using this spectral characteristic in remote sensing satellites can effectively capture the characteristics of the earth surface. Based on the color information of the object, we input the optical sequence signal with a specific wavelength to VOFET-DR, which corresponds to the electromagnetic wave signal reflected by the satellite receiving the object. Due to the distributed reservoir states of the device, the sequence signal can be mapped to the reservoir state space of different dimensions according to the temporal characteristics of the specific wavelength, thereby separating the physical image information of the landform into different feature channels, which is more conducive to identification, as shown in Fig. 4b.

Next, the three electromagnetic spectral features of Ultraviolet (UV), Visible (Vis), Near Infrared (NIR) of the image are respectively input into the grouped-RC system as independent characteristic channels, as shown in Fig. 4c. The NIR characteristic channel of the image is taken as an example. First, the NIR electromagnetic band image is preprocessed including cropping, binarization and resizing, and rejoining to a $16 \times 100$ pixels image. Correspondingly, the reservoir consists of 4 parallel sub-reservoirs, each of which has a different $V_{GS}$ to give the sub-reservoir different spatiotemporal characteristics. Each sub-reservoir includes 100 VOFET-DR physical nodes, and generates 100 feature outputs for feature fusion and training of the output layer, as shown in Supplementary Information Fig. S26. The relationship between the mathematical model and the physical hardware for this grouped-RC is shown in Supporting Information Fig. S27. Finally, by inputting images of different feature channels of UV, Vis, and NIR into the grouped-RC network, performing feature fusion and training, the feature recognition of satellite remote sensing images is successfully confirmed in Fig. 4d, and their recognition accuracy exceeds 95%. In addition, Fig. 4e shows the effect of adding different numbers of sub-reservoirs with different spatiotemporal characteristics on the recognition accuracy. It can be found that the recognition accuracy reaches 94.9% after adding 4 sub-reservoirs with different spatiotemporal characteristics, which shows that the rich reservoir state space is more conducive to separating the spatiotemporal characteristics of the signal and improving the recognition accuracy. Furthermore, the task is also performed using single-layer, double-layer Artificial Neural Network (ANN) networks and Convolutional Neural Networks (CNN). The accuracy achieved is 88.1%, 96.1% and 92.1% respectively, which, verified that the grouped-RC is comparable to the traditional mainstream technology in term of recognition efficiency. Notably, the grouped-RC only requires weight training for part of the RC connected to the output layer, resulting in a significantly lower number of weights (-2400) compared to single-layer ANN (-1,440,288), double-layer ANN (-2,884,896) and CNN (-25,600) as depicted in Fig. 4f. This demonstrates that the grouped-RC achieves comparable accuracy while reducing the weight-related computational cost by over 90% compared to ANNs and CNNs.

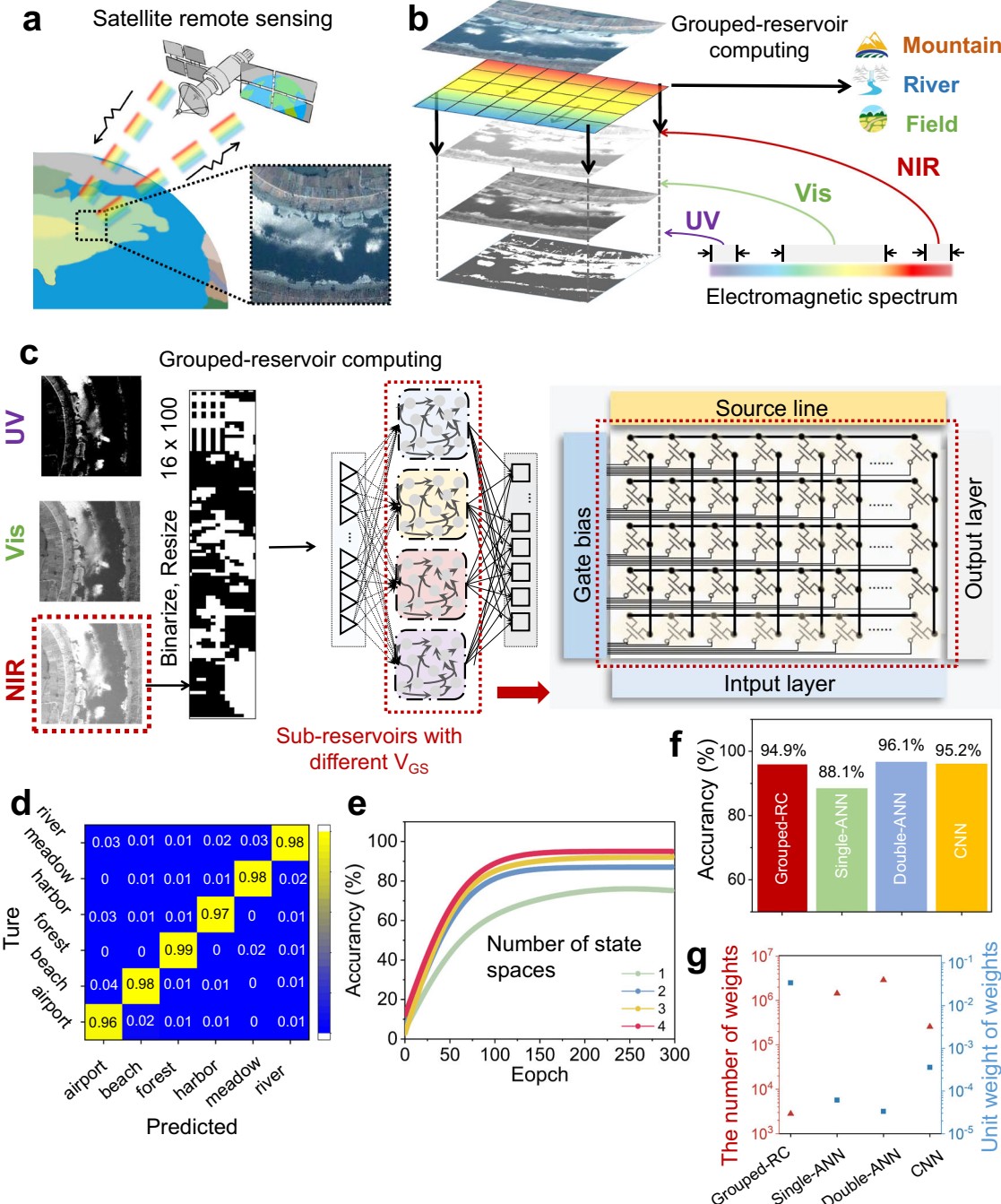

**Fig. 4 | Satellite remote sensing image recognition based on grouped-RC.**
**a** Schematic diagram of satellite remote sensing image monitoring. **b** Schematic
diagram of remote sensing image recognition based on grouped-RC. **c** After pre-
processing different feature channels of remote sensing images, they are input to
reservoir nodes with different biases (0 V, −3 V, −8 V, −10 V), and feature outputs are
generated respectively for training of output layers. **d** Recognition accuracy of six
classic scenes. **e** The influence of the number of sub-reservoirs with different
dynamic states on the identification accuracy. **f** Comparison of the accuracy of
group-RC with single-layer ANN, two-layer ANN and CNN. **g** The number of weight
and unit weight efficiency of grouped-RC with single-layer, two-layer ANN and CNN.

## Traffic trajectory prediction

As a time-series signal task, traffic trajectory prediction is an important
application of machine vision, and the different spatiotemporal char-
acteristic signals it contains have become a challenge for accurate
prediction. For example, as shown in the traffic scene described in
Fig. 5a, in a traffic road, due to the different motion rates of different
traffic elements (pedestrians, bicycles, cars), objects generate different
time frame and space frame information flows. Therefore, by using
VOFET-DR with extensive spatiotemporal properties as a physical node
to effectively extract the spatiotemporal features of moving object
signals, we expect to achieve accurate traffic trajectory prediction.

First, for object motion trajectory detection, we combine the
light-induced short-term memory effect of the device with the optical
flow method and the inter-frame difference method to obtain the
motion rate (time frame information) and orientation (spatial frame
information) of the moving object respectively. As shown in Fig. 5b, a
reservoir matrix consisting of N × M pixel modules $P_{(n, m)}$ is used to
receive the dynamic information flow, where each pixel consists of a
VOFET-DR and acts as a single reservoir to receive the sequence signal $i$
$(t_n)$ of the dynamic information flow. N and M depend on the resolu-
tion of the image to fit the image, i.e. N and M are the image length and
width, respectively. Due to the light-induced short-term memory effect

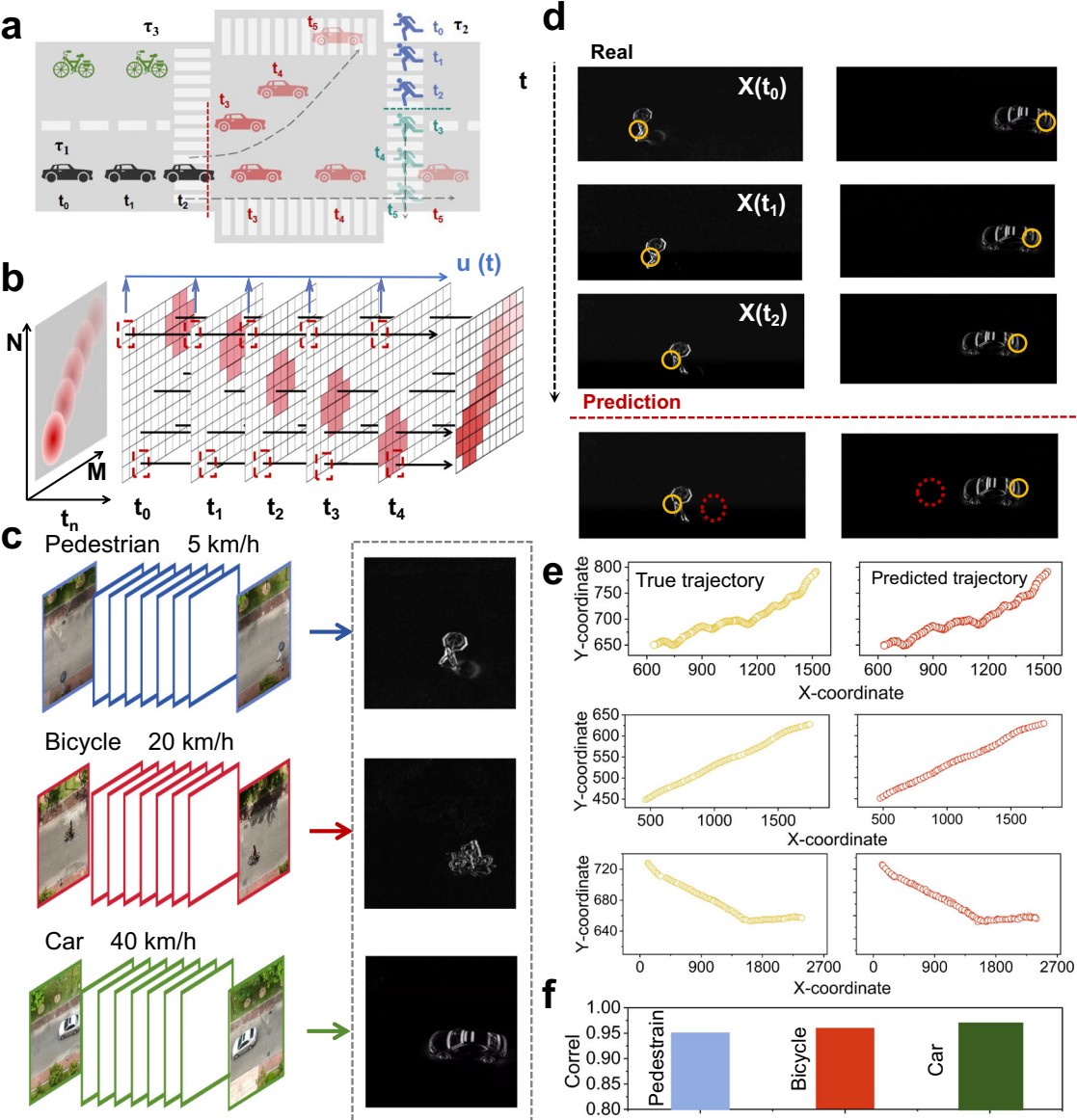

**Fig. 5 | Traffic trajectory prediction. a** Schematic diagram of traffic trajectory prediction. Different traffic elements have different spatiotemporal characteristics. **b** VOFET-DR arrays for mapping spatiotemporal vision information. Pixels in specific columns of a time frame ($t_0$ to $t_4$) form a time visual sequence u (t). **c** The spatial frame information of different traffic elements. **d** The velocity and

coordinate information of the object in the previous three frames ($X(t_0)$, $X(t_1)$, $X(t_2)$) as input and output to train the encoding system. **e** The prediction of motion trajectories for pedestrians, bicycles, and cars. **f** The correlation coefficients between the predicted trajectories and the actual trajectories of pedestrian, bicycle, and car.

of VOFET-DR, the current memory generated by the moving object (red ball) at the previous time position gradually decreases with time, and forms a strength difference with the current memory at the current time position. Thus, the dynamic information flow from $t_0$ to $t_4$, the trajectory of the object through the current memory layer will be formed having a direction, and the trajectory of the object can be identified according to the layer component. It is worth noting that the current gradient generated by the difference in electric current memory at different times depends on the object velocity. Therefore, by utilizing the gradient component of the current memory, the object velocity can be further calculated. Furthermore, due to the different velocities of various traffic elements, it is possible to adapt to different traffic elements by adjusting the gate voltage of VOFET-DR and setting its time characteristics. Finally, by comparing the current memory of VOFET-DR at different time intervals, i.e., frame difference method, the spatial frame information of the moving object can be obtained based on the difference result, as shown in Supplementary Information

Fig. S28 and Fig. 5c show the spatial frame information of different traffic elements. Furthermore, utilizing the obtained velocity and orientation information mentioned above, the encoding system built by the VOFET-DR reservoir layer predicts the future frames of the moving objects, as shown in Supplementary Information Fig. S29. When the system detects a moving object, as shown in Fig. 5d, the system uses the velocity and coordinate information of the object in the previous three frames ($X(t_0)$, $X(t_1)$, $X(t_2)$) as input and output to train the encoding system. A well-trained system will then make predictions for the velocity and coordinates of the next frame, marked with a red circle. Figure 5e presents the prediction of motion trajectories for pedestrians, bicycles, and cars, and it reveals a good overlap between the predicted and actual trajectories. In Fig. 5f, the correlation coefficients (0.951, 0.959, 0.969) between the predicted trajectories and the actual trajectories of pedestrians, bicycles, and cars are respectively displayed, indicating accurate trajectory prediction results.

Overall, in this study, we first analyze the relationship between the computing capacity of physical reservoir computing and the richness of reservoir states in reservoir devices based on the mathematical model of the reservoir. It is reveals that the computing capacity of physical reservoir computing is heavily dependent on two physical coefficients of reservoir devices: the feedback intensity (F) and the time characteristic ($\tau$) (Details are discussed in Supplementary Information Note 1). To achieve high-performance reservoir computing, it is crucial to enhance the dynamic range of the F and $\tau$ at the physical device level. The dynamic range of the F is influenced by the photo-generated charge transport modulation mechanism and transport efficiency. Meanwhile, the vertical transistor architecture with nanoscale transport distance can not only improve the transport efficiency of photogenerated charges but also provide richer modulation by the additional terminal. Additionally, the non-uniform gate field can precisely regulate charge transport. This overcomes the limitations of traditional optoelectronic memristors and lateral transistors in terms of single modulation mechanism and low charge transport efficiency, which narrows the scale of feedback intensity. Meanwhile, the scale of the $\tau$ is greatly affected by the energy state of trapped charges. Therefore, inducing photogenerated charges with different energy states by using light pulses with different wavelengths can effectively expand the time scale of the device.

Finally, the key advantages of VOFET-DR are summarized: (i) Distributed reservoir state space and grouped-reservoir. VOFET-DR overcomes the limitation of a monotonic reservoir state space in shallow-RC, a major challenge in RC. By leveraging the distributed reservoir states of VOFET-DR and combining multiple physical mechanisms, the carrier dynamics can be adaptively adjusted, resulting in a reservoir state space with 1152 states. This enables the mapping of sequential signals using different carrier dynamics, achieving grouped-RC. (ii) Ultra-wide range ratio of spatial and temporal characteristics. The narrow spatial and temporal characteristics scale of shallow-RC caused by monotonic physics mechanisms (a major problem with traditional neuromorphic devices) are solved by VOFET-DR. The device exhibits a wide range ratio on spatial (650) and temporal characteristics (2640), both of which outperform the reported neuromorphic device for RC. Hence, the grouped-RC network exhibit high precision in both image recognition (94%) and dynamic prediction (95%). (iii) Negligible additional power density. VOFET-DR can achieve a significant increase in feedback intensity (650 times), but only generates an additional negligible power density (approximately $10^{-4}$ mJ s$^{-1}$ cm$^{-2}$).

## Discussion

In summary, we for the first time introduce an innovative organic vertical neuromorphic transistor with distributed reservoir states, specifically designed for grouped-RC. By coupling multivariate physics mechanisms to enrich carrier dynamics, our proposed VOFET-DR as a reservoir exhibits a distributed reservoir state space with 1152 reservoir states, overcoming the limitations of traditional shallow-RC with monotonic reservoir state space in achieving high-precision recognition and prediction in complex spatiotemporal tasks. The device exhibits ultra-wide range rates (2640 and 650) in both temporal and spatial characteristics, outperforming the reported neuromorphic reservoir devices. Further, the grouped-RC network implemented based on the device can simultaneously demonstrate over 94% recognition accuracy and 95% prediction correlation in tasks of different spatiotemporal types, respectively. Moreover, at the level of approximate recognition accuracy, the number of weights used by this grouped-RC network is reduced by over 90% compared to mainstream ANN and CNN network architectures. Therefore, this work provides an innovative strategy for developing high-performance reservoir computing networks and devices suitable for different types of spatiotemporal tasks, and has great potential in the development of advanced artificial intelligence computing.

## Methods

### Materials

The P0FDIID organic semiconductor is synthesized according to previous work[46]. N2200 are purchased from Derthon Optoelectric Materials Science Technology Co. Ltd. and use without further purification. P0FDIID and N2200 were dissolved in chlorobenzene at a concentration of 5 mg ml$^{-1}$, and the N2200 solution (25 wt%) is then mixed into the P0FDIID solution as the active layer. The electrolyte material polyvinyl alcohol (PVA) (Mw = 67 kDa) and DL-malic acid (Mw = 134.09) are both purchased from Aladdin Biochemical Technology Co., Ltd. MXene (5 mg/ml in deionized water) is purchased from XFnano Materials Tech Co., Ltd. and further diluted to 3 mg/ml as the source electrode of device.

### Fabrication of VOFET-DR

The Si/SiO$_2$ (300 nm) substrates are first cleaned with acetone and then sonicated in isopropanol, trichloromethane and deionized water in turn for 5 min and eventually dry with N$_2$ gas. Afterwards, the cleaned substrate is treated with plasma for 6 mins. Immediately, PVA (dissolved in a mixed solution of deionized water and absolute ethanol 60 wt%: 40 wt%) is spin-coated on SiO$_2$ (3000 rpm, 40 s) and annealed at 100 °C (10 min) in a nitrogen atmosphere. After that, a 1 nm thick Al$_2$O$_3$ film is deposited on the PVA surface by atomic deposition technique. The 50 nm gold source is thermally evaporated onto the MXene film through a shadow mask as the pin of the source electrode (2–3 nm), which is convenient for connecting the probe during testing. Then, in a nitrogen glovebox, the mixed solution (P0FDIID:N2200) is spin-coated on the sample at 1250 rpm for 60 s and then placed at 150 °C for 10 min for evaporation of residual solution to form the active layer (-65 nm). Finally, 50 nm gold is thermally evaporated onto the PDVT−10 film through a shadow mask as the drain electrode. The effective channel area (200 μm × 200 μm) is determined by the overlapping area between the MXene and top gold drain electrode.

**Optoelectronic measurement.** The electrical and synaptic performance is characterized by the semiconductor parameter analyzer (Keysight B2902A and Keysight 4200-SCS). The AFM (Nanoscope III, Veeco Instruments, Inc.) is used to measure the mixing films morphology under ambient conditions. UV−vis absorption spectra is recorded to characterize ultraviolet-visible-near infrared spectrophotometer (Shimadzu UV-3600 Plus). The SEM images of MXene are obtained on a focusion beam/SEM (Nova NanoSEM 230). KPFM measurements are performed in ambient air using a Bruker Fastscan AFM instrument.

### Network training

The hyperspectral image training dataset in satellite remote sensing images is derived from Hyperspectral Remote Sensing Scenes, where 10 different scenarios are selected for each landform type. Hyperspectral images are converted into binary gray images by image processing and input into the device in the form of light pulses according to the corresponding gray release coding. Devices with different gate biases ($V_{GS}$ = 0 V, -3V, -8V, −10V) have different carrier dynamics states to act as different sub-reservoirs. The outputs of all sub-reservoirs are input in parallel to the input layers of the fully connected network, the network size is 400 × 6. The fully connected network is trained by the MATLAB Deeplearning Toolbox, utilizing the Softmax output function and the logistic regression to supervise the learning.

## Data availability

The data that support the plots within these paper and other findings of this study are available from the corresponding authors upon request.

## Code availability

The code that supports the theoretical plots within this paper is available from the corresponding author upon request.

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

## Acknowledgements

The authors are grateful for financial support from National Natural Science Foundation of China (62374033, U21A20497) and Fujian Science & Technology Innovation Laboratory for Optoelectronic Information of China (2021ZZ129).

## Author contributions

H.C. and C.G. conceived the project. C.G. is responsible for device fabrication, data acquisition and analysis, device mechanism analysis and physical field simulation calculation, algorithm design, and paper writing. D.L., C.X., W.X., and X.Z. are responsible for algorithm design and verification. C.Z. is responsible for device mechanism analysis and algorithm verification. J.B. is responsible for the synthesis of organic semiconductors. Z.L., Y.H., and T.G. are responsible for overseeing the overall work. H.C. supervises the project. Everyone has read the manuscript.

## Competing interests

The authors declare no competing interests.
