## [Peer Review File · Nature Communications]

REVIEWER COMMENTS

Reviewer #1 (Remarks to the Author):

In this article, the authors report on a vertical neuromorphic transistor with distributed reservoir states, which is achieved by combining the vertical structure field-effect transistor with an organic heterojunction semiconductor material system that has a broad-spectral persistent photoconductive effect. The coupling of multivariate physics expands the amount of reservoir state space in the reservoir, thereby increasing its computing capacity. The results show that the device achieves an impressive large-scale temporal and spatial characteristic and shows excellent results in both static and dynamic information processing based on reservoir computing. This article is well-written, the viewpoints are innovative, and the research content is rich, covering a wealth of device performance research, physical field simulation calculation, and application verification. I hope the following suggestions can help the authors improve the manuscript during revision. I would recommend the publication after minor revisions.

Suggestions:

1. In this device, according to the explanation of the working mechanism, the PPC effect comes from the organic semiconductor heterojunction. It is well known that in this bulk heterojunction, the mixing ratio of p-type and n-type semiconductor materials directly affects the electrical characteristics of the semiconductor. What I want to know is whether, as a dynamic memory or a synaptic device, the different proportions of n-type and p-type have a significant impact on the memory effect. Does it have a significant impact on its application in reservoir computing. Additionally, please clarify the performance requirements of this dynamic memory device for reservoir computing.
2. Regarding the structure of the device, I noticed that the author used PVA as a passivation layer to reduce the effect of the off current on the device. However, why did the author add a layer of Al₂O₃, and what is its purpose? In addition, the device has also been confirmed to have short-term memory current behavior induced by electrical pulses. Is it associated with the use of PVA?
3. In Figure 3b, the author calculates the activation energy of electrons under different physical programming conditions. However, no relevant calculation instructions are provided. Please add the relevant method instructions.
4. In the process of clarifying the working mechanism, the author used the physics field calculation of COMSOL to help analyze the working mechanism of the device, which is a very interesting tool. Can the author provide the relevant parameters of his simulation calculation?
5. In Figure 5, the author describes the capture of dynamic objects by means of frame difference. How is this applied to the device to incorporate this method, and how is its velocity and trajectory information obtained?
6. There are some tense errors in the article. Some are in the present tense, and some are in the past tense. The author is requested to pay attention to modifying the spelling and grammar problems in the article.

Reviewer #2 (Remarks to the Author):

In the present paper, the authors introduce a vertical transistor structure serving as a physical reservoir with distributed reservoir states. The paper describes an impressive development starting from the physics and optoelectronics of the vertical device structure over the synaptic properties of the device & device integration to the evaluation of the reservoir and benchmarking vs. other network structures. The paper describes certainly a very timely and interesting development in the field of neuromorphic computing and using physical reservoirs as sensing units. However, I cannot recommend the paper for publication in Nature Communication in its current form as its rich and versatile content is at the same time its biggest weakness. The authors spent an enormous amount of work on analyzing their devices and explaining all effects, however, it makes the paper almost unreadable due to the very high density of content. At the same time though, fairly simple but decisive questions e.g., about the device structure and why the authors have chosen e.g., these particular materials, thickness, configuration of trapping layers, etc. are not addressed. So there is no learning for me as a reader for example about the structure-property-relationship of the vertical devices and what needs to be changed in order to further improve the rich dynamics of the systems. I am not questioning the results of the authors but the only thing I see in the paper is a very nice and well-executed demonstration of a system - but I do not learn much about how to design it.

In my opinion, the authors should revise the paper in a way that they focus on the structure-property relation of the vertical devices and how they are connected to the properties of the reservoir nodes. A demonstration of a classification or prediction task is needed - but probably one demonstration would be sufficient. Furthermore, many claims and comparisons done in the paper are rather questionable, e.g., "vertical architecture employed greatly increases the feedback strength of the device while causing negligible additional power." I do not see any measurable evidence for that. or "Notably, the weight training cost is reduced by 98% compared to traditional double-layer artificial neural networks (ANNs)." -> Of course a double-layer ANN cannot outperform a reservoir but a double-layer ANN is simply not what you should compare with.

Reviewer #3 (Remarks to the Author):

The manuscript "Toward grouped-reservoir computing: ultra-short channel organic neuromorphic vertical transistor with distributed reservoir states for efficient recognition and prediction" with Changsong Gao et al. demonstrated a vertical neuromorphic transistor consisting of photon artificial synapse with organic vertical FET. The device utilized the evolving states of different artificial synapse memory currents to map sequence signals in multiple dimensions, thereby capturing the temporal and spatial characteristics of sequence signals more comprehensively, which is suitable for the requirements of grouped hardware reservoir computing. In particular, due to the ultra-short channel of the vertical transistor, it can effectively facilitate the separation and capture of photo-generated charges. By inputting broadband (ultraviolet to near-infrared) light pulses, the device exhibits rich memory states and demonstrated good performance in both image recognition and prediction of dynamic information. Overall, the topic of this work is truly interesting. The manuscript is well organized. After revising some questions, I recommend the publication of this work.

Q1. For vertical field-effect transistors, the roughness of the meshed source-drain region can easily cause stability issues. Additionally, the use of a semiconductor mixed in the active layer can further contribute to film roughness, exacerbating the problem. How was this situation addressed in the experiment? What was the yield rate of the devices?

Q2. For this system, in addition to POFDIID as a donor, is it applicable to other materials? I think this is an important issue related to the universality of the scheme. In addition, what factors need to be considered to realize the artificial photosynaptic device of this system?

Q3. In reservoir computing, reservoirs require lateral connections to enable the representation of temporal context. How did the authors achieve this process?

Q4. As there are multiple variants of reservoir networks, it would be useful to include the mathematical model of the reservoir network topology.

Reviewer #1 (Remarks to the Author):**Comment:**

In this article, the authors report on a vertical neuromorphic transistor with distributed reservoir states, which is achieved by combining the vertical structure field-effect transistor with an organic heterojunction semiconductor material system that has a broad-spectral persistent photoconductive effect. The coupling of multivariate physics expands the amount of reservoir state space in the reservoir, thereby increasing its computing capacity. The results show that the device achieves an impressive large-scale temporal and spatial characteristic and shows excellent results in both static and dynamic information processing based on reservoir computing. This article is well-written, the viewpoints are innovative, and the research content is rich, covering a wealth of device performance research, physical field simulation calculation, and application verification. I hope the following suggestions can help the authors improve the manuscript during revision. I would recommend the publication after minor revisions.

Response:

We greatly appreciate your review and recognition. We are making revisions and additions to the manuscript based on your suggestions one by one. We always strive to provide high-quality manuscripts and have made necessary modifications and supplements under your guidance. Thank you very much for your professional opinions and suggestions, which are invaluable to our research work. We will continue to work hard to ensure that our manuscript is more comprehensive and accurate. Thank you again for your support and encouragement!

Suggestion 1:

In this device, according to the explanation of the working mechanism, the PPC effect comes from the organic semiconductor heterojunction. It is well known that in this bulk heterojunction, the mixing ratio of p-type and n-type semiconductor materials directly affects the electrical characteristics of the semiconductor. What I want to know is whether, as a dynamic memory or a synaptic device, the different proportions of n-type and p-type have a significant impact on the memory effect. Does it have a significant impact on its application in reservoir computing. Additionally, please clarify the performance requirements of this dynamic memory device for reservoir computing.

Reply to Suggestion 1:

We greatly appreciate your meaningful question. We agree with your observation that the ratio of p-type semiconductors to n-type semiconductors in the bulk heterojunction (BHJ) system has a significant impact on the memory effect of the device.

In our previous research work, we investigated the effect of different mixing

ratios on the memory current of the device, as shown in **Figure R1**. Based on different p/n semiconductors blending ratios, BHJ is used as the transistor channel, and the device is subjected to the same optical pulse. As shown in **Figure R1.1a**, at a low n-type blending ratio, the device exhibits better memory retention characteristics. However, as the n-type blending ratio increases, as shown in **Figures R1.1b** and **R1.1c**, the memory effect of the device gradually decreases, transitioning from long-term memory to short-term memory. For reservoir computing, devices serving as physical nodes in the reservoir layer need to possess short-term memory characteristics to meet the recursive requirements. If long-term memory characteristics are used, although the recursive condition can be satisfied, it is difficult to map the input signal to high-dimensional space due to its lack of nonlinearity. Therefore, after experimental optimization, a blending ratio of 20% wt (N2200) is adopted in this work.

Figure R1.1 The impact of different blending ratios on the memory characteristics induced by optical pulses in the device (450nm, pulse width $\Delta t = 100$ ms, $V_{DS} = -1$ V, light intensity $P_{in} = 0.01\text{mW/cm}^2$). **a** The blending ratio of N2200 is 5% wt. **b** The blending ratio of N2200 is 10% wt. **c** The blending ratio of N2200 is 25% wt.

In addition, based on the suggestions, we have summarized the requirements of reservoir computing for semiconductor devices with dynamic memory characteristics:

1. Characteristics of short-term memory.

It is well known that reservoir computing is a special form of recurrent neural network, so the short-term memory effect enables devices to equivalently implement neural networks with recursive connections, which is a component of reservoir state characteristics (or echo state characteristics). When a device with short-term memory characteristics is used as a reservoir, the state of the reservoir network will be determined by the input of the device and the real-time electrical state of the device, that is, the reservoir can be asymptotically stabilized, so that the reservoir can exhibit good performance on the synchronous timing signal. In this work, the device has optical pulse-induced short-term memory characteristics, which meets the requirements of this application.

2. Nonlinearity.

In a reservoir computing network, the reservoir layer mainly consists of a network of randomly interconnected nodes, which allows the input signal to be mapped non-linearly into the space of high-dimensional states. Due to the increase of

feature differentiation, the input signal that is difficult to separate in the low-dimensional space becomes separable. Therefore, as a physical node in the reservoir, the device itself needs to possess nonlinear dynamic behavior as a nonlinear function that maps real-world time signals. In this work, the output current of the device increases nonlinearly after being stimulated by the light pulse. After the light pulse ends, the output current decreases nonlinearly, so it has the application conditions of RC.

3. A large number of reservoir states.

In reservoir computing, the processing ability of timing signal depends greatly on the high-dimensional reservoir space state of the reservoir. In the process of the signal being input into the reservoir, the reservoir device projects different timing signals to different reservoir states according to the nonlinear time characteristics based on the nonlinear and short-term memory effects, and then predicts the data of the timing signals combined with linear separation. Therefore, enough reservoir states can effectively map different timing signals into a high-dimensional space. Based on the PPC effect induced by broad-spectrum light stimulation and vertical structure, the device can achieve 1152 reservoir states with a wide range of temporal characteristics, meeting the application requirements of RC.

Revised:

- 1) In the Supplementary Information file, we add a detailed discussion of the relationship between reservoir computing and physical devices as a new **Supplementary Information Note 1** to clarify the properties required for physical devices applied to reservoir computing.
- 2) We have added a new **Supplementary Information Note 5** to the Supplementary Information to discuss the impact of different mixing ratios on device-related characteristics.

Suggestion 2:

Regarding the structure of the device, I noticed that the author used PVA as a passivation layer to reduce the effect of the off current on the device. However, why did the author add a layer of Al₂O₃, and what is its purpose? In addition, the device has also been confirmed to have short-term memory current behavior induced by electrical pulses. Is it associated with the use of PVA?

Reply to suggestion 2:

This is a very meaningful question. In the experiment, PVA is dissolved in a mixture of ultrapure water and anhydrous ethanol, while the solvent for MXene, which serves as the meshed source electrode, also uses ultrapure water. To prevent mutual dissolution between PVA and MXene, a thin layer of Al₂O₃ film is added between them via atomic layer deposition.

In addition, the PVA film contributes to the voltage pulse-induced short-term memory current behavior, as shown in **Figure R1.2**. This is because PVA contains a large number of hydroxyl groups that can capture electrons, leading to

voltage-induced short-term memory current behavior. This phenomenon is consistent with previously reported works. Due to the wider temporal and spatial characteristics of the short-term memory current induced by optical pulses in this study, the focus of this research is not on the voltage-induced short-term memory current.

Figure R1.2 The transistor device in this study consists of a pure POFDIID semiconductor as the channel and PVA as the charge-capturing layer.

Suggestion 3:

In Figure 3b, the author calculates the activation energy of electrons under different physical programming conditions. However, no relevant calculation instructions are provided. Please add the relevant method instructions.

Reply to suggestion 3:

We greatly appreciate the constructive suggestions provided by the reviewer. We will addition this part in **Supplementary Note 4**.

For charge trapping, the decay constant τ is functionally related to the activation energy E_a and the ambient temperature T . It can be described by the following equation^{R1}:

$$\tau = \tau_0 \exp\left(\frac{E_a}{k_B T}\right)$$

where τ_0 and k_B represent the thermal constant and the Boltzmann constant, respectively. E_a is typical the activation energy for charge trapping. According to the Arrhenius equation, the activation energy is inversely proportional to the ambient temperature T . As the ambient temperature increases, the activation energy will decrease. Conversely, as the ambient temperature decreases, the activation energy will increase. This is because high temperatures increase the average kinetic energy of molecules, making it easier to overcome the energy barrier between reactants, thereby reducing the activation energy. Low temperatures slow down the movement of molecules, making it more difficult to overcome the energy barrier, resulting in

increased activation energy. Therefore, changes in temperature can trigger changes in the decay constant τ , as shown in **Figure R1.3**. Since the value of τ can be directly obtained from the decay of I_{DS} , the corresponding activation energy E_a can be calculated by testing the I_{DS} induced by light pulses of specific wavelengths at different ambient temperatures.

Figure R1.3 Current decay varies with ambient temperature.

Revised:

We have added a new **Supplementary Information Note 4** to the Supplementary Information to clarify the calculation of activation energy.

Suggestion 4:

In the process of clarifying the working mechanism, the author used the physics field calculation of COMSOL to help analyze the working mechanism of the device, which is a very interesting tool. Can the author provide the relevant parameters of his simulation calculation?

Reply to suggestion 4:

We greatly appreciate your advice.

For the calculation of potential distribution, it is mainly based on the equations $E = -\nabla V$ and $\rho_v = \nabla \cdot (\epsilon_0 \epsilon_r E)$, where the relative permittivity of the material is a key parameter. The simulation parameters of COMSOL are supplemented in the **Supplementary Information Table 4**.

	X	Y	Relative permittivity
Gate (Si)	500 nm	50 nm	-
Dielectric layer (SiO ₂)	500 nm	100 nm	4.5
Passivation layer (PVA)	500nm	20 nm	3.5
Source (MXene)	75 nm	3 nm	-
Active layer	500 nm	100 nm	2.5
Drain (Au)	500 nm	50 nm	-

Supplementary Information Table 4. COMSOL related simulation parameters.

Revised:

We have added a new **Supplementary Information Table 4** to the Supplementary Information file to demonstrate the relevant simulation parameters.

Suggestion 5:

In Figure 5, the author describes the capture of dynamic objects by means of frame difference. How is this applied to the device to incorporate this method, and how is its velocity and trajectory information obtained?

Reply to suggestion 5:

For the detection of moving objects, we mainly rely on the frame difference method combined with the light-responsive current of the device to achieve it. Firstly, the main algorithm of frame difference method is as follows:

$$O(t)_{(n,m)} = \begin{cases} 0, & F(t) - F(t-1) \leq \Delta_{th} \\ 1, & F(t) - F(t-1) > \Delta_{th} \end{cases}$$

Where $O(t)_{(n,m)}$ represents the binary output state of the motion information at position (n,m) in the t-th frame. $F(t)$ represents the sub-pixel output at position (n,m) in the t-th frame. Δ_{th} represents the threshold. By subtracting the output states of corresponding pixels in the previous and next frames, if it is a stationary object, the difference will be smaller than the difference threshold. If it is a moving object, the difference will exceed the difference threshold.

For a photoelectric sensor, its photoelectric response formula is $P_{in}R = I_{out}$, where P_{in} (light intensity) can be considered as the input information of the sub-pixel, which represents the motion status information of the object. I_{out} (current) can be considered as the output state of the sub-pixel. Therefore, a segment of motion information can be regarded as an information flow composed of multiple frames of images. Due to the different distributions of P_{in} received by each sub-pixel in each frame, we refer to it as the input matrix distribution $X_{(n,m)}$. As a result, the output state distribution of the sensor array I_{out} is generated with corresponding spatial state distribution due to the P_{in} distribution, which we refer to as the output matrix distribution $Y_{(n,m)}$. Due to the light-induced short-term memory current effect of VOFET-DM, its current magnitude depends on the intensity of the input light pulse. Therefore, we use the output current I_{DS} of the device and its first derivative with respect to time t as the output $Y_{(n,m)}$, to better reflect the differences in light intensity between each frame of the image. Therefore, in the static case where the P_{in} distribution of the previous frame is the same as the P_{in} distribution of the next frame, the input matrix distribution $X(t-1)_{(n,m)} = X(t)_{(n,m)}$, and as a result, the output matrix distribution of the sensor array I_{out} will

also be the same, $Y(t-1)_{(n,m)} = Y(t)_{(n,m)}$. Combining the frame differencing method mentioned above, we can consider $Y(t-1)$ as equivalent to $F(t-1)$ and $Y(t)$ as equivalent to $F(t)$. As a result, in this case, the motion detection output will be 0, indicating black. Similarly, in the case of motion, where the P_{in} distribution of the previous frame is different from the P_{in} distribution of the next frame, the input matrix distribution $X(t-1)_{(n,m)} \neq X(t)_{(n,m)}$, and as a result, the output matrix distribution of the sensor array I_{out} will also be different, $Y(t-1)_{(n,m)} \neq Y(t)_{(n,m)}$. Combining the frame differencing method mentioned above, we can consider $Y(t-1)$ as equivalent to $F(t-1)$ and $Y(t)$ as equivalent to $F(t)$. As a result, in this case, the motion detection output will be 1, indicating white.

Regarding the motion speed of objects, as mentioned above, the output current state I_{out} of the device already represents the spatial distribution of the image. Therefore, we can represent the sub-pixel output $I_{out}(x, y, t)$ in the t -th frame as the value of sub-pixel $I_{out}(x, y)$ at time t . After a time interval of dt , this pixel has moved by (dx, dy) in the next frame. Since these pixels are the same and their intensities remain unchanged, we can represent this as:

$$I(x, y, t) = I(x + dx, y + dy, t + dt)$$

We can perform a Taylor expansion on the expression as follows:

$$I(x + dx, y + dy, t + dt) = I(x, y, t) + \frac{\partial I}{\partial x} dx + \frac{\partial I}{\partial y} dy + \frac{\partial I}{\partial t} dt$$

By combining the two equations, we can obtain:

$$\frac{\partial I}{\partial x} \frac{dx}{dt} + \frac{\partial I}{\partial y} \frac{dy}{dt} + \frac{\partial I}{\partial t} \frac{dt}{dt} = 0$$

Where $\frac{dx}{dt}$ and $\frac{dy}{dt}$ represent the displacements in the x and y dimensions respectively, dt represents the time interval. The velocities in the x and y dimensions are defined as V_x and V_y respectively. Additionally, $\frac{\partial I}{\partial x}$, $\frac{\partial I}{\partial y}$, and $\frac{\partial I}{\partial t}$ represent the partial derivatives of the sub-pixel current state with respect to the x , y , and t dimensions, respectively. These derivatives can be directly obtained by taking the corresponding first-order derivatives of the $I_{DS}(t)$. By solving these equations, we can calculate the velocity information of the object in the x and y dimensions.

Suggestion 6:

There are some tense errors in the article. Some are in the present tense, and some are in the past tense. The author is requested to pay attention to modifying the spelling and grammar problems in the article.

Reply to suggestion 6:

We really appreciate your meticulous review of our work. We will carefully review and correct the language issues of the article.

Revised:

All tenses in the Results and Experiment section have been modified.

Reference

[R1] Flynn, J. H.; Wall, L. A. A quick, direct method for the determination of activation energy from thermogravimetric data. *J. Polym. Sci. Part B: Polym. Lett.* **1966**, 4, 323–328.

Reviewer #2 (Remarks to the Author):**Comment:**

In the present paper, the authors introduce a vertical transistor structure serving as a physical reservoir with distributed reservoir states. The paper describes an impressive development starting from the physics and optoelectronics of the vertical device structure over the synaptic properties of the device & device integration to the evaluation of the reservoir and benchmarking vs other network structures. The paper describes certainly a very timely and interesting development in the field of neuromorphic computing and using physical reservoirs as sensing units.

Response:

Thank you very much for your review and affirmation of our work and for your valuable suggestions. We greatly value your comments and suggestions, which are essential to this work. I will ensure that they are incorporated into subsequent revisions and improvements of the article to improve the quality and accuracy of the paper.

Comment:

However, I cannot recommend the paper for publication in Nature Communication in its current form as its rich and versatile content is at the same time its biggest weakness. The authors spent an enormous amount of work on analyzing their devices and explaining all effects, however, it makes the paper almost unreadable due to the very high density of content.

Response:

Thank you very much for the reviewer's suggestions.

As the reservoir devices used in reservoir computing need to exhibit **nonlinear response characteristics, short-term memory characteristics, and a large number of reservoir states**, it is necessary to systematically investigate the performance of

devices. The feedback intensity F of the nonlinear response characteristics and the time characteristics τ of short-term memory have a significant impact on the richness of reservoir states. Therefore, our research incorporates many important data, which is crucial for readers to better understand our research.

Of course, we appreciate the suggestions made by the reviewers. To provide a more organized presentation of the research content, we have rearranged the dense research material in the paper and added subheadings. This helps readers to better comprehend our research.

The study of Figure 2 in the manuscript is refined into four parts:

2.1. Field effect characteristics of the device. (including Figure 2b to 2c).

2.2. Nonlinear response and short-term memory characteristics of the device. (including Figures 2d to 2g).

2.3. Nonlinear mapping of multi-bit signals. (including Figures 2h to 2i).

2.4. Distributed reservoir states of VOFET-DR. (including Figures 2j to 2o).

Revised:

The second part adds sub-headings according to the research content.

2.1. Field effect characteristics of the device.

2.2. Nonlinear response and short-term memory characteristics of the device.

2.3. Nonlinear mapping of multi-bit signals.

2.4. Distributed reservoir states of VOFET-DR. Among them, the original third part is changed to **2.4.**

Comment:

At the same time though, fairly simple but decisive questions e.g., about the device structure and why the authors have chosen e.g., these particular materials, thickness, configuration of trapping layers, etc. are not addressed. So there is no learning for me as a reader for example about the structure-property-relationship of the vertical devices and what needs to be changed in order to further improve the rich dynamics of the systems. I am not questioning the results of the authors but the only thing I see in the paper is a very nice and well-executed demonstration of a system - but I do not learn much about how to design it. In my opinion, the authors should revise the paper in a way that they focus on the structure-property relation of the vertical devices and how they are connected to the properties of the reservoir nodes.

Response:

We greatly appreciate the reviewer's suggestions regarding the current weakness of the article, and we further refine the logic and content of the paper to make it easier for you and other readers to understand the research content of this work.

In this study, **we first analyze the relationship between the computing capacity** of physical reservoir computing and **the richness of reservoir states** in reservoir devices **based on the mathematical model of the reservoir**. It is concluded that the computing capacity of physical reservoir computing depends significantly on two physical coefficients of reservoir devices: **the feedback intensity (F) and the**

time characteristic (τ). Therefore, it is crucial to enhance the dynamic range of the F and τ at the physical device level for high-performance reservoir computing.

The dynamic range of the F is greatly influenced by the photogenerated charge transport modulation mechanism and transport efficiency. Meanwhile, the vertical transistor architecture with nanoscale transport distance can not only improve the transport efficiency of photogenerated charges but also provide richer modulation by the additional terminal. Moreover, the non-uniform gate field can precisely regulate charge transport. This compensates for the limitations of traditional optoelectronic memristors and lateral transistors in terms of single modulation mechanism and low charge transport efficiency, which narrow the scale of feedback strength. On the other hand, **the scale of the τ is greatly affected by the energy state of trapped charges.** Therefore, inducing optoelectronic charges with different energy states by using light pulses with different wavelengths can effectively expand the time scale of the device.

In summary, we use **organic semiconductor materials with broad spectral absorption characteristics to achieve a large-scale τ and couple it with the vertical architecture to broaden the F range of the reservoir.** As a result, the richness of reservoir states in the reservoir devices is greatly enhanced, leading to an improved computing capacity.

Furthermore, to ensure that readers can better understand the relationship between reservoir computing and physical devices, as well as our research efforts, we proceed from the bottom-up approach, systematically discussing the following three parts in detail: **1. Reservoir computing model, 2. Physical coefficients of the devices, and 3. Design strategies for physical reservoir devices.** We organize and supplement this discussion in **Supplementary Information Note 1**, and make revisions to the relevant content in the manuscript based on the suggestions of the reviewers.

1. Reservoir computing model

First, as shown in **Figure R2.1**, the reservoir computing network is a Recurrent Neural Network model based on a delayed feedback system. The core idea is to process the input signal through a fixed, sparse, randomly connected recurrent network (called a reservoir), where the node needs to receive the input signal and its own feedback signal, and then update its state according to a pre-set connection weight and nonlinear activation function. After that, the state of the reservoir is mapped to the output layer by linear output weights, and the timing signal is predicted by a simple algorithm of the output layer. Therefore, the relationship between its input I and output O can be described by the following mathematical equation:

$$\begin{aligned}x(t + 1) &= f(W_{\text{res}} * x(t) + W_{\text{in}} * u(t)) \\y(t) &= W_{\text{out}} * x(t)\end{aligned}$$

Where $x(t)$ is the network state at the current moment. $x(t + 1)$ is the network state at the next moment. $u(t)$ is the input at the current moment. $y(t)$ is the output at the

current moment. W_{res} is the recursive weight matrix of the reservoir, which is used to control the update of the current network state, W_{in} is the input weight matrix, which is used to control the effect of the input signal on the state vector. W_{out} is the output weight matrix, which is used to map the state vector of the reservoir to the output. $f()$ represents a nonlinear function.

Figure R2.1. Model diagram of reservoir computing network based on delayed feedback system.

Therefore, in a reservoir computing network based on a delayed feedback system, multiple reservoir nodes are required to feed back their own output as input to satisfy the recursive condition, which means that the output of the reservoir node will directly or indirectly affect its own input. So, in physical reservoir computing, this requires the physical nodes to be able to physically respond, transmit, and memorize signals, expressed as a series of node self-feedback. This can be described by the following equation:

$$\frac{dx(t)}{dt} = f(t, x(t), x(t - \tau))$$

$$\theta = \tau/N$$

Where τ is the duration of the delay and N is the number of nodes and the θ is the time-step and $f()$ is a system function that depends on the physical system.

It can be found that in order to satisfy the mathematical architecture of RC at the physical device level, the output states of the physical devices must satisfy: **nonlinear response characteristics, short-term memory characteristics and a large number of reservoir states**. Among them, the number of reservoir states, as nodes, determines the computational capacity of reservoir computing, and the number depends greatly on the former two.

2. Physical coefficients of the devices

1) **For nonlinear response characteristics**, as shown in **Figure R2.2** and in Figures 2d and 2e of the manuscript, the device generates nonlinear response behavior induced by light pulses or voltage. By fitting its mathematical expression as follows:

$$I = I_{(t-1)} + A \left[1 - \frac{\exp(-(t - t_0))}{\tau} \right]$$

$$A \sim f(F)$$

Where $I_{(t-1)}$ is the initial current state, the A coefficient is the difference between $I_{(t-1)}$ and $I_{(\infty)}$, which belongs to the intrinsic characteristics of the device and depends on the feedback intensity F of the device.

The mathematical equation describes the evolution process of the reservoir state after the physical reservoir is fed a signal, which can be regarded as a reservoir state space. Therefore, it can be found that there is a significant relationship between the number of reservoir state spaces and A coefficient. In order to make the reservoir have as many state spaces as possible to provide as many reservoir states as possible, as shown in the **Figure R2.3a**, the A coefficient needs to be able to be regulated in a large range.

2) On the other hand, **for the short-term memory characteristics**, the mathematical expression it is fitted:

$$I(t) = I_{spike} + D_1 \exp(-t/\tau_1) + D_2 \exp(-t/\tau_2)$$

Where, τ_1 and τ_2 represent the characteristic time of the fast decay and slow decay process, respectively. D_1 and D_2 represent the prefactor. I_{spike} represent the current constant. Since the physical nodes in the reservoir need to satisfy the point-wise separation property of the reservoir, in other words, be able to respond significantly to different sequence signals, we use the characteristic time τ_1 of the device to better evaluate the mapping ability of the device to the sequence signal. Therefore, the decay process of the memory current can be seen as:

$$I(t) = I_{spike} + D_1 \exp(-t/\tau)$$

The time characteristic τ determines the decay rate of the memory current, which directly affects the evolution process of the reservoir state, the feedback relationship between the input $I(t-1)$ and output $I(t)$ of the signal in the reservoir state space. In order to make the reservoir have more reservoir state space to provide as many reservoir states as possible, as shown in the **Figure R2.3b**, the time characteristic τ needs to be adjustable on a large scale.

Therefore, based on the above analysis, we need to meet the large-scale feedback intensity F and time characteristics τ for the design of the device to provide as many reservoir states as possible.

Figure R2.2 Nonlinear function extracted from the light response curve (I-t) of the device.

Figure R2.3 a Different reservoir states due to different feedback intensities. **b** Different reservoir states due to different temporal characteristics.

3. Design strategies for physical reservoir devices

Since the dynamic range of A coefficient is related to the feedback intensity F of the device, **the problem of the dynamic range of A coefficient can be solved by increasing F**. In the process of current rise, the relationship between the change amount of photocurrent ΔI and the input light intensity can be regarded as macroscopically:

$$\frac{\Delta I}{P_{in}} = \frac{I_{spike} - I_0}{P_{in}} = F$$

Where scaling factor F, the feedback strength, shows the weight between the output and the input of the device.

Currently, optoelectronic devices used in reservoir computing mainly focus on dynamic optoelectronic memory resistors. **However, due to the limited ports, it is difficult to further adjust the physical mechanism of the device or the carrier**

transmission process, resulting in a relatively fixed dynamic range of the feedback intensity F. As a result, the output of the device is almost determined by the most recent input, making the device itself a relatively fixed nonlinear transformation function, thereby limiting its ability to provide a more diverse reservoir state space to the system.

After the introduction of field effect characteristics, the gate voltage at the third terminal can more accurately regulate the distribution of carriers in the photoelectric field effect transistor, thus realizing a richer carrier transport process. Therefore, using field effect transistors to design reservoirs is greatly beneficial to improve the dynamic range of the feedback intensity F to create more reservoir states. However, **for traditional transistors,** the carriers in the channel are transported in the horizontal direction. **The micro-level long transport distance results in severe charge loss of the carriers due to interface defects, bulk defects, and so on during the transport process.** This greatly limits the carrier transport efficiency and gate control ability, thereby affecting the feedback intensity of the device.

And for the **new vertical architecture transistor,** it not only has the basic function of gate electric field modulation, but also due to its source and drain being located at the bottom and top of the semiconductor layer respectively, carriers are influenced by the electric field for transport in the vertical direction. This means that the transport distance is only determined by the thickness of the active layer (which is ~65 nm in this work). This nano-level channel length significantly reduces the carrier transport distance and reduces charge loss during the transport process, thereby improving carrier transport efficiency. In addition, the field distribution between the source and drain electrodes under gate control is non-uniform. The region near the source electrode will experience stronger gate control. The non-uniform distribution of the adjustable electric field is beneficial for regulating the separation balance between holes and electrons inside optoelectronic device, reducing the recombination caused by different electron-hole transmission rates, and enhancing the gate control capability of the device. As a result, **the dynamic range of the feedback intensity F can be effectively improved.**

On the other hand, the short-term memory characteristics of the device are essentially determined by the charge trapping and de-trapping processes in the channel. **The energy state of the trapped charge greatly affects the process of de-trapping, resulting in time characteristics τ at different scales.** For the mechanism of charge trapping, there is a functional relationship between the time characteristic τ and the charge activation energy E_a and the ambient temperature T , which can be described by the following physical equations:

$$\tau = \tau_0 \exp\left(\frac{E_a}{k_B T}\right)$$

In optoelectronic devices, photogenerated charges are generated by the photoelectric effect, and photons of different energies can excite photogenerated electron-hole pairs at different energy levels. **Photogenerated electron-hole pairs at**

different energy levels have different activation energies, which is the minimum energy required to form a photogenerated charge. Therefore, under room temperature conditions, we choose an organic semiconductor (POFDIID) with broad-spectrum absorption characteristics as the channel material for the device, with an absorption spectrum ranging from 300 nm to 1000 nm. By exciting photo-generated electrons in different energy states through photons with different frequencies, we can provide varying activation energies, as shown in the **Figure R2.4**. Next, by introducing an electron acceptor organic semiconductor (N2200) with energy levels that match, a PN heterojunction system is formed. The potential barrier generated at the interface can trap and de-trap photo-generated electrons with different activation energies during the transport process, resulting in a diverse time characteristic.

Figure R2.4 Charge activation energy generated by light pulses of different wavelengths.

Therefore, in this device, we have chosen the organic semiconductor POFDIID, which has a broad spectral absorption characteristic, as the channel material. It serves as both the charge transport layer and the light-absorbing layer. The broad spectral absorption range (300nm-1000nm) of POFDIID enables the device to exhibit a wide temporal characteristic. The underlying reasons will be explained in the subsequent discussion. At the same time, considering that N2200 has good solubility in common organic solvents, and has a high electron affinity. After blending N2200 with POFDIID they can form a suitable energy band relationship, which is conducive to trapping and storing charges. Therefore, in order to achieve short-term memory characteristics, we mixed a small amount of N2200 organic semiconductor in the channel as a charge trapping site-point. MXene is used as mesh source electrode due to its excellent conductivity. In the experiment, the thickness of the active layer (mixed film) is ~65 nm, and the MXene nanosheet served as the mesh source electrode, with a thickness of 2~3 nm (as shown in the **Figure R2.5**).

In summary, we use organic semiconductor materials with broad spectral absorption characteristics to achieve a large-scale τ and couple it with the vertical architecture to broaden the F range of the reservoir. As a result, the richness of reservoir states in the reservoir devices is greatly enhanced, leading to an improved computing capacity.

Figure R2.5. **a** Height map of MXene mesh source electrode tested by atomic force microscopy. **b** Thickness of film at positions 1, 2 and 3. **c** Thickness of active layer.

Revised:

1) The subtitle of the first part of the **Results** section of the manuscript has been revised: **1. Grouped-RC and device design.**

2) Line 12 of the **Results** is revised as follows:

“However, achieving this process at the physical device level is a challenge, as it requires reservoir devices to possess device attributes of non-linear response characteristics and short-term memory characteristics, while also needing a wide dynamic range of feedback intensity and time characteristics to meet the demands of a large number of reservoir states. (Details are discussed in **Supplementary Information Note 1**). Although the use of dynamic memristors has been widely reported, its limited number of terminals can easily cause the reservoir to become a relatively fixed nonlinear function²⁰. At the same time, the limitations of the photogenerated charge transport efficiency due to the long transport distance of conventional transistors can easily lead to a narrow range of feedback intensities F .”

3) We summarize this part of the discussion at the end of the **Results** section to clarify the relationship between reservoir computing and physical devices in our study

“Overall, in this study, we first analyze the relationship between the computing capacity of physical reservoir computing and the richness of reservoir states in reservoir devices based on the mathematical model of the reservoir. It is revealed that the computing capacity of physical reservoir computing is heavily dependent on two physical coefficients of reservoir devices: the feedback intensity (F) and the time characteristic (τ) (Details are discussed in **Supplementary Information Note 1**). To achieve high-performance reservoir computing, it is crucial to enhance the dynamic range of the F and τ at the physical device level. The dynamic range of the F is influenced by the photogenerated charge transport modulation mechanism and transport efficiency. Meanwhile, the vertical transistor architecture with nanoscale transport distance can not only improve the transport efficiency of photogenerated charges but also provide richer modulation by the additional terminal. Additionally,

the non-uniform gate field can precisely regulate charge transport. This overcomes the limitations of traditional optoelectronic memristors and lateral transistors in terms of single modulation mechanism and low charge transport efficiency, which narrow the scale of feedback intensity. Meanwhile, the scale of the τ is greatly affected by the energy state of trapped charges. Therefore, inducing photogenerated charges with different energy states by using light pulses with different wavelengths can effectively expand the time scale of the device.

4) In the supplementary information file, we add a detailed discussion of the relationship between reservoir computing and physical devices as a new **Supplementary Information Note 1**.

5) Add a new Figure S7 to the Supplementary Information:

Supplementary Information Fig. S7 | Height image of MXene. **a** Height map of MXene mesh source electrode tested by atomic force microscopy. **b** Thickness of film at positions 1, 2 and 3.

Comment:

A demonstration of a classification or prediction task is needed - but probably one demonstration would be sufficient.

Response:

We would like to express our gratitude to the reviewers for their valuable suggestions.

The validation of these two different types of tasks is crucial for objectively evaluating the performance of reservoir computing and they depend on different parameters. The task of static image classification focuses more on the range of feedback intensity F , while the prediction of dynamic time series signal focuses more on temporal characteristics τ . Previous reports usually can only perform well in one task, while our work can perform well on both tasks, which is one of the superiority of our work.

In the process of static image classification, the original images are input into the reservoir in binary encoded format, which leads to the reservoir receiving a large and complex sequence of signals. At this stage, the narrow dynamic range of feedback intensity can lead to the reservoir associating different sequence signals with the same category during the process of image feature extraction, resulting in a decrease in classification accuracy. Therefore, it pays more attention to the dynamic range of feedback intensity. **In the feature extraction process of time series signal prediction**, dynamic time series signals with rich temporal features need to be accompanied by an appropriate time characteristic τ . For example, high-frequency time series signals require a small time characteristic τ to adapt and avoid excessive sensitivity caused by a large τ , which would lead to different time series being associated with the same task. On the other hand, low-frequency time series signals require a larger time characteristic τ to adapt and avoid the loss of historical information in the reservoir, which would break the recursive condition. Therefore, it emphasizes more on the dynamic range of time characteristic τ .

Comment:

Furthermore, many claims and comparisons done in the paper are rather questionable, e.g., "vertical architecture employed greatly increases the feedback strength of the device while causing negligible additional power." I do not see any measurable evidence for that.

Response:

We greatly appreciate the suggestions of reviewer.

In this device, the significant enhancement of feedback intensity is attributed to the precise modulation of channel carrier transport by the gate voltage. The gate and drain terminals of the transistor are blocked and disconnected by the insulating layer, so the power cost ($V_{GS} \times I_{GS}$) generated by it is extremely small, even negligible.

As shown in **Figure R2.6a** and Figures 2e and 2g in the manuscript, when the gate voltage is floating, the device exhibits short-term memory current behavior after being input with a light pulse, with a feedback intensity 50 ($\text{nA mW}^{-1} \text{cm}^{-2}$). When an additional gate voltage bias of -15 V is applied, the feedback intensity increases to 9.53 ($\mu\text{A mW}^{-1} \text{cm}^{-2}$), as shown in Figure 2k in the manuscript, which is an increase of 19000%. Specifically, the P_{gs} ($V_{GS} \times I_{GS}$) density generated by the additional bias is only $10^{-4} \sim 10^{-3}$ ($\text{mJ s}^{-1} \text{cm}^{-2}$), as shown in **Figure R2.6b** and Supplementary Information Figure 14, which accounts for only 0.0004% of the I_{ds} density 25 ($\text{mJ s}^{-1} \text{cm}^{-2}$). Its power density can be considered negligible.

Figure R2.6 a Devices with different gate voltage biases are input with a light pulse, resulting in short-term memory current. (Pulse width: 100 ms, wavelength: 650 nm, $V_{DS} = -1$ V, $P_{in} = 0.01$ mW/cm²). **b** The gate voltage generates leakage current.

Revised:

1) Line 13 of section 2.4 of the manuscript is amended as follows:

“However, an important consideration is the additional power caused by gate control, which is largely dependent on the gate leak current (I_{gs}). As shown in **Supplemental Information Fig. S13**, the P_{gs} ($V_{GS} \times I_{gs}$) density is approximately $10^{-4} \sim 10^{-3}$ (mJ s⁻¹cm⁻²), which accounts for only 0.0004% of the I_{ds} density 25 (mJ s⁻¹ cm⁻²). At the same time, when an additional gate voltage bias of -15V is applied, the feedback intensity increases to 9.53(μ A mW⁻¹ cm⁻²), which is a 190-fold increase compared to no gate (50 nA mW⁻¹ cm⁻²). Therefore, the power derived from external electric field could be negligible.”

2) The last sentence in the **Results** section should be modified as follows:

(iii) **Negligible additional power density.** VOFET-DR can achieve a significant increase in feedback intensity (650 times), but only generates an additional negligible power density (approximately 10^{-4} mJ s⁻¹ cm⁻²).

Comment:

"Notably, the weight training cost is reduced by 98% compared to traditional double-layer artificial neural networks (ANNs)."  Of course a double-layer ANN cannot outperform a reservoir but a double-layer ANN is simply not what you should compare with.

Response:

We are very grateful for the suggestions.

We agree with the reviewers' suggestions and have removed the relevant statements from the manuscript accordingly.

In addition, we have additionally supplemented the case of convolutional neural networks (CNNs) architectures to facilitate more objective comparisons.

Revised:

1) These few sentences in the summary, introduction and conclusion section are deleted.

“Notably, the weight training cost is reduced by 98% compared to traditional

double-layer artificial neural networks (ANNs).”

“Moreover, under the condition of achieving a similar level of recognition accuracy, the number of weights used in the grouped-RC in this work is reduced by over 98% compared to the double-layer ANN.”

“Moreover, under the condition of achieving a similar level of recognition accuracy, the number of weights used in the grouped-RC in this work is reduced by over 98% compared to the double-layer ANN.”

1) In line 32 of Part 4 of the manuscript, make the following changes:

“Furthermore, the task is also performed using single-layer, double-layer Artificial Neural Network (ANN) networks and Convolutional Neural Networks (CNN). The accuracy achieved is 88.1%, 96.1% and 95.2% respectively, which, verified that the grouped-RC is comparable to the traditional mainstream technology in term of recognition efficiency. Notably, the grouped-RC only requires weight training for part of the RC connected to the output layer, resulting in a significantly lower number of weights (~2400) compared to single-layer ANN (~1440288), double-layer ANN (~2884896) and CNN (~256000) as depicted in **Fig. 4f**. This demonstrates that the grouped-RC achieves comparable accuracy while reducing the weight-related computational cost by over 90% compared to ANNs and CNNs.”

3) Replace Figures 4f and 4g with:

Figure 4f

Figure 4g

Reviewer #3 (Remarks to the Author):

The manuscript “Toward grouped-reservoir computing: ultra-short channel organic neuromorphic vertical transistor with distributed reservoir states for efficient recognition and prediction” with Changsong Gao et al. demonstrated a vertical neuromorphic transistor consisting of photon artificial synapse with organic vertical FET. The device utilized the evolving states of different artificial synapse memory currents to map sequence signals in multiple dimensions, thereby capturing the temporal and spatial characteristics of sequence signals more comprehensively, which is suitable for the requirements of grouped hardware reservoir computing. In particular, due to the ultra-short channel of the vertical transistor, it can effectively facilitate the separation and capture of photo-generated charges. By inputting broadband (ultraviolet to near-infrared) light pulses, the device exhibits rich memory states and demonstrated good performance in both image recognition and prediction of dynamic information. Overall, the topic of this work is truly interesting. The manuscript is well organized. After revising some questions, I recommend the publication of this work.

Response:

We greatly appreciate your review and recognition. We are making revisions and additions to the manuscript based on your suggestions one by one. We always strive to provide high-quality manuscripts and have made necessary modifications and supplements under your guidance. Thank you very much for your professional opinions and suggestions, which are invaluable to our research work. We will continue to work hard to ensure that our manuscript is more comprehensive and accurate. Thank you again for your support and encouragement!

Q1:

For vertical field-effect transistors, the roughness of the meshed source-drain region can easily cause stability issues. Additionally, the use of a semiconductor mixed in the active layer can further contribute to film roughness, exacerbating the problem. How was this situation addressed in the experiment? What was the yield rate of the devices?

Replay to the Q1:

We fully agree with the reviewer's opinion and thanks for raising this important issue. The reviewer is right, and the source roughness directly affects the stability of the vertical transistor. In this work, the concentration parameters of the MXene solution are optimized. When the concentration of the MXene solution is low (0.1 mg/ml), the conductivity of the MXene film is poor, and it is difficult to control the injection of carriers in the channel, causing the device to easily fail to turn on. When the MXene solution concentration is high (1 mg/ml), the roughness of the film is very large, and the rougher source surface easily penetrates the active layer and connects the drain electrode, causing the device to be prone to conduction and short circuit.

Although the addition of a certain proportion of N2200 in P0FDIID can affect the transfer characteristics of the transistor, as shown in **Figure R3.1**, such as sub-threshold swing and threshold voltage, the devices can still demonstrate transistor performance even at low N-type proportions (>40%). In the experiment, we measured 125 devices, of which 115 devices have good transistor performance. The transfer curve is shown in **Figure R3.2**. The success rate of the fabricated devices can reach 90%.

Figure R3.1 The transfer characteristics of the device after mixing P0FDIID with different proportions of N2200 as the active layer.

Figure R3.2 Among 125 vertical organic transistors manufactured, 115 have good transistor performance.

Q2:

For this system, in addition to P0FDIID as a donor, is it applicable to other materials? I think this is an important issue related to the universality of the scheme. In addition, what factors need to be considered to realize the artificial photosynaptic device of this system?

Reply to Q2:

Thank you very much for the suggestions of the reviewers. We will supplement this part in Supplementary Information Note 5.

Other p-type semiconductors can also serve as donors to meet the requirements of

reservoir computing for physical devices in terms of short-term memory and non-linear response characteristics, provided they meet the matching conditions. As shown in **Figure R3.1**, we can still achieve the PPC effect on devices based on the PDVT-10 : N2200 heterojunction. This mechanism based on charge trapping in semiconductors is universal and has been previously reported in literature.

The energy band alignment and the mixing ratio between p-type and n-type materials need to be considered. we investigated the effect of different mixing ratios on the memory current of the device, as shown in **Figure R3.1**. Based on different p/n semiconductors blending ratios, BHJ is used as the transistor channel, and the device is subjected to the same optical pulse. As shown in **Figure R3.1a**, at a low n-type blending ratio, the device exhibits better memory retention characteristics. However, as the n-type blending ratio increases, as shown in **Figures R3.1b** and **R3.1c**, the memory effect of the device gradually decreases, transitioning from long-term memory to short-term memory. For reservoir computing, devices serving as physical nodes in the reservoir layer need to possess short-term memory characteristics to meet the recursive requirements. If long-term memory characteristics are used, although the recursive condition can be satisfied, it is difficult to map the input signal to high-dimensional space due to its lack of nonlinearity. Therefore, after experimental optimization, a blending ratio of 20% wt (N2200) is adopted in this work.

Figure R3.1. Persistent photoconductivity effect observed in devices with PDVT-10 : N2200 (10%wt) mixed film as the active layer under optical pulse stimulation (310 nm Pulse width: 100 ms, P_{in} : $0.01\text{mW}/\text{cm}^2$, $V_{DS} = -1\text{ V}$, $V_{GS} = 0\text{ V}$).

Figure R3.2 The impact of different blending ratios on the memory characteristics induced by optical pulses in the device (450nm, pulse width $\Delta t = 100\text{ms}$).

100 ms, $V_{DS} = -1$ V, light intensity $P_{in} = 0.01\text{mW/cm}^2$). **a** The blending ratio of N2200 is 5% wt. **b** The blending ratio of N2200 is 10% wt. **c** The blending ratio of N2200 is 25% wt.

Revised:

1) We have added a new **Supplementary Information Note 5** to the Supplementary Information to discuss the impact of different mixing ratios on device-related characteristics.

Q3:

In reservoir computing, reservoirs require lateral connections to enable the representation of temporal context. How did the authors achieve this process?

Reply to Q3:

We greatly appreciate the suggestions from the reviewers.

In this physics reservoir computing system, the schematic diagram of its lateral connections is shown in Figure R3.3. The reservoir state is related to the input sequence pulses and the real-time electrical state of the reservoir. There are lateral chain connections between nodes in each reservoir, which can be demonstrated by the real-time pulse response of the I-t relationship curve. For example, in the input sequence pulses '100011', '101011', and '110101', the last bit is 1 for all three, but the reservoir states are different. If there were no lateral connections, the reservoir states would be closer to the same output state.

Figure R3.3. The devices were individually input with different sequences of multi-bit optical pulse signals, demonstrating distinct current states.

Q4:

As there are multiple variants of reservoir networks, it would be useful to include the mathematical model of the reservoir network topology.

Reply to Q4:

We greatly appreciate the reviewer's questions. To ensure that readers have a better understanding of the relationship between reservoir computing and physical devices, as well as our research efforts, we will discuss the model, organize and supplement this discussion in **Supplementary Information Note 1**, and modify the relevant content in the manuscript based on the suggestions of the reviewers.

As shown in **Figure R3.4**, the reservoir computing network is a type of recursive neural network model based on a delayed feedback system. Its core idea is to process input signals through a fixed, sparse, randomly connected recurrent network called the reservoir. In this network, nodes receive input signals and their own feedback signals, and then update their states based on pre-set connection weights and non-linear activation functions. Afterwards, the state of the reservoir is mapped to the output layer using linear output weights, and a simple algorithm in the output layer is used to predict the temporal signal. Therefore, the relationship between the input I and the output O can be described by the following mathematical equation:

$$\begin{aligned}x(t+1) &= f(W_{res} * x(t) + W_{in} * u(t)) \\ y(t) &= W_{out} * x(t)\end{aligned}$$

In this model, $x(t)$ represents the current network state, $x(t+1)$ represents the next network state. $u(t)$ represents the current input. $y(t)$ represents the current output. W_{res} is the recursive weight matrix of the reservoir, used to control the update of the current network state. W_{in} is the input weight matrix, which used to control the influence of input signals on the state vector. W_{out} is the output weight matrix, used to map the state vector of the reservoir to the output. $f()$ represents a non-linear function.

Therefore, in order to enable recursion in the network, delayed feedback is required. In physical reservoir computing, physical nodes need to physically connect, transmit and respond to delayed signals, resulting in a series of virtual node behaviors. This can be described by the following equation:

$$\begin{aligned}\frac{dx(t)}{dt} &= F(t, x(t), x(t-\tau)) \\ \theta &= \tau/N\end{aligned}$$

Where τ is the duration of the delay, N is the number of nodes, θ is the time step, and $F()$ is a system function that depends on the physical system. Therefore, it can be observed that at the physical device level, meeting the mathematical architecture requirements of RC involves semiconductor devices having output states that exhibit non-linear response characteristics, short-term memory characteristics, and a large number of reservoir states. The number of reservoir states is highly dependent on the first two requirements.

For non-linear response, as shown in **Figure R3.5** and Figures 2d and 2e in the manuscript, the device exhibits non-linear response behavior induced by voltage pulses or light pulses. This behavior can be mathematically expressed by fitting the data as follows:

$$I = I_{(t-1)} + A \left[1 - \frac{\exp(-(t - t_0))}{\tau} \right]$$

Where $I_{(t-1)}$ represents the initial current state, A is the difference between $I(\infty)$ and $I_{(t-1)}$, which is an intrinsic characteristic of the device and represents the feedback intensity, and τ is the rise time characteristic. This mathematical equation describes the evolution process of the reservoir state in the physical reservoir after being influenced by the input signals. It can be viewed as a reservoir state space. Therefore, it can be observed that there is a significant relationship between the number of reservoir state spaces and the feedback intensity A . In order to have as many reservoir state spaces as possible and to provide a wide range of reservoir states, it is necessary for the feedback intensity A to be adjustable within a large range.

On the other hand, for the short-term memory characteristic, as shown in Figure R11, it can be mathematically expressed by fitting the data as follows:

$$I(t) = I_{spike} + D_1 \exp(-t/\tau_1) + D_2 \exp(-t/\tau_2)$$

Where, τ_1 and τ_2 represent the characteristic time of the fast decay and slow decay process, respectively. D_1 and D_2 represent the prefactor. I_{spike} represent the current constant. Since the physical nodes in the reservoir need to satisfy the point-wise separation property of the reservoir, in other words, be able to respond significantly to different sequence signals, the characteristic time τ_1 of the device to better evaluate the mapping ability of the device to the sequence signal. Therefore, the decay process of this memory current can be seen as:

$$I(t) = I_{spike} + D_1 \exp(-t/\tau)$$

Where τ is the time characteristic of the device, which is an intrinsic parameter of the device. It determines the decay rate of the memory current, thus directly affecting the state of the reservoir, i.e., the nonlinear dynamics on which the evolution of signals in the reservoir state space relies. Therefore, in order to have more reservoir state space and provide as many reservoir states as possible, it is necessary for the time characteristic τ to be adjustable within a larger range. Based on the analysis above, we have constructed an architecture for grouped reservoir computing, as shown in **Figure R3.6**. By controlling the gate voltage and the wavelength of the input light, different time characteristics τ and feedback intensity $F(A)$ can be set, enabling the system to have multiple reservoir state spaces. These spaces are updated with different τ and F functions to update the states of the virtual nodes. These nodes are set at the trailing edge of the pulses, with intervals determined by the pulse width.

Figure R3.4. Model diagram of reservoir computing network based on delayed feedback system.

Figure R3.5 Nonlinear function extracted from the light response curve (I-t) of the device.

Figure R3.6. Schematic diagram of the grouped reservoir network as a demonstration of our work.

Revised:

1) Line 12 of the **Results** is revised as follows:

“However, achieving this process at the physical device level is a challenge, requiring reservoir devices to simultaneously have a wide dynamic range of feedback intensity and temporal characteristics to meet the needs of a large number of reservoir states (details are discussed in **Supplementary Information Note 1**). Although the use of dynamic memristors has been widely reported, its limited number of terminals can easily cause the reservoir to become a relatively fixed nonlinear function²⁰. At the same time, the limitations of the photogenerated charge transport efficiency due to the long transport distance of conventional transistors can easily lead to a narrow range of feedback intensities F .”

2) In the Supplementary Information file, we add a detailed discussion of the relationship between reservoir computing and physical devices as a new **Supplementary Information Note 1**.

REVIEWERS' COMMENTS

Reviewer #1 (Remarks to the Author):

The authors successfully addressed the comments. I recommend to publish the manuscript.

Reviewer #2 (Remarks to the Author):

I would like to thank the authors for the careful and open revision of the paper. I think the readability of the paper has been significantly improved. In particular, SI Note 1 helps the reader a lot to connect the device properties to the important measures of the reservoir performance. Also the other revisions concerning the comparison to other ANNs are fair. I still believe that the paper is difficult to read as it is so content rich and requires knowledge from very different fields. However, this is no judgement on the quality of the work and hence, I am happy to recommend this manuscript for publication.

Reviewer #3 (Remarks to the Author):

All of my concerns have been properly addressed and I would like to recommend the acceptance of the revised manuscript in its current version.